# EHR2Path: Comprehensive Pathway-Level Modeling of Longitudinal Patient Trajectories from Multimodal Electronic Health Records

## Abstract

Forecasting how a patient's condition is likely to evolve, including possible deterioration, recovery, treatment needs, and care transitions, could support more proactive and personalized care, but requires modeling heterogeneous and longitudinal electronic health record (EHR) data. Yet, existing approaches typically focus on isolated prediction tasks, narrow feature spaces, or short context windows, limiting their ability to model full patient pathways. To address this gap, we introduce EHR2Path, a multimodal framework for forecasting and simulating full in-hospital patient pathways from routine EHRs. EHR2Path converts diverse clinical inputs into a unified temporal representation, enabling modeling of a substantially broader set of patient information, including radiology reports, physician notes, vital signs, medication and laboratory patterns, and dense bedside charting. To support long clinical histories and broad feature spaces, we introduce a Masked Summarization Bottleneck that compresses long-term history into compact, task-optimized summary tokens while preserving recent context, improving both performance and token efficiency. In retrospective experiments on MIMIC-IV, EHR2Path enables next-step pathway forecasting and iterative simulation of complete in-hospital trajectories, while outperforming strong baselines on directly comparable tasks. These results demonstrate the feasibility of pathway-level modeling from routine EHRs, and indicate potential for supporting anticipatory clinical decision-making. We will release our code upon acceptance.

## 1 Introduction

Anticipating how patients evolve during a hospital stay remains a central challenge in data-driven clinical decision support. Hospital care depends on repeated decisions about monitoring, treatments, transfer, and discharge, all of which require anticipating how an individual patient's condition is likely to evolve. EHRs capture structured diagnosis and treatment codes, continuous physiologic measurements, medication administrations, bedside observations and free-text notes, offering a detailed but highly heterogeneous view of patient trajectories. Patient pathways (Richter & Schlieter, 2019) span multiple temporal scales, from long-term disease progression over a lifetime to short-term trajectories within a single healthcare episode. Here, we focus on full in-hospital patient pathways, as visualized in fig. 1, which are documented in detail in EHRs and represent a critical window for clinical intervention. Unlike models that predict a single downstream outcome, a comprehensive in-hospital patient pathway model could forecast how a patient's full state is likely to evolve over time, enabling anticipation of deteriorating vital signs, adverse events, treatment needs and care transitions, and thereby supporting more proactive and personalized care. Yet translating these heterogeneous records into pathway-level forecasts remains difficult. To address this challenge, we propose EHR2Path, a method for forecasting and simulating full in-hospital patient pathways from routine EHRs.

Substantial progress has been made in machine learning for EHRs, with a large body of work focusing on *outcome prediction*, targeting individual downstream endpoints such as mortality, length of stay, diagnosis prediction, or specific laboratory values. Another line of work moves toward *trajectory modeling*, aiming to represent or generate sequences of patient events over time. While both directions have produced important

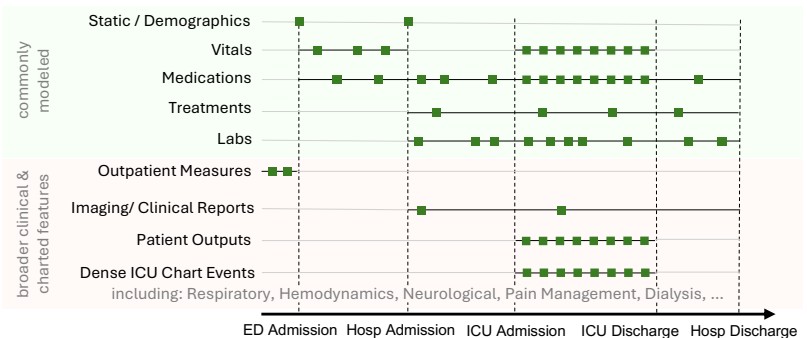

Figure 1: Short-term Patient Pathway visualization within Emergency Department, Hospital and ICU.

advances, current methods often operate on restricted subsets of the EHR, limited context windows, or narrowly defined tasks. As a result, they typically do not model the full multimodal, longitudinal patient state at pathway level, and they rarely evaluate iterative simulation of complete in-hospital trajectories.

Recent advances in large language models (LLMs) provide a promising foundation for pathway-level EHR modeling. Their key advantage in this setting is the ability to represent highly heterogeneous clinical information, including free text, numerical measurements, and categorical events, within a unified autoregressive modeling framework while preserving temporal and contextual structure. By serializing diverse clinical signals into a shared sequence space, LLMs offer a flexible alternative to modality-specific pipelines and narrowly scoped task models. Prior work has shown that such models can learn rich representations from clinical text (Yang et al., 2022) and can support unified modeling of diverse EHR signals (Hur et al., 2023; McDermott et al., 2023; Makarov et al., 2025). However, directly applying LLMs to full hospital trajectories remains challenging, since clinically relevant information is distributed across long, dense, and irregular histories that quickly exceed practical context limits.

To address current limitations, we introduce **EHR2Path**, a pathway-centric approach for forecasting and simulating full in-hospital patient pathways from routine EHRs. EHR2Path restructures multimodal hospital data, including structured events, laboratory measurements, medications, chart data, and clinical notes, into a unified, structured textual representation that preserves temporal structure within and across hospital stages (ED, ward, ICU). EHR2Path makes three key advances: (1) it performs *pathway-level state forecasting and simulation* by generating the next-hour EHR state over a broad set of clinical events and physiological values, rather than targeting a narrow endpoint; (2) it unifies the full breadth of EHR inputs, including structured, continuous, and free-text data, across ED, ward, and ICU in a single trajectory model; and (3) it introduces a *Masked Summarization Bottleneck* that restricts access to extended history through a small set of learned, section-specific summary tokens, while preserving recent events as uncompressed text, enabling efficient use of long clinical context. Given a patient's past trajectory, EHR2Path predicts the next hour's sparse EHR state and can be rolled out iteratively to simulate future pathways.

In a retrospective evaluation on the MIMIC-IV dataset (Johnson et al., 2023b), we show that EHR2Path improves next-hour forecasting and multi-day simulation, outperforming baselines in both forecasting and outcome prediction tasks while using context more efficiently. Together, these results establish a path toward pathway-level modeling of heterogeneous in-hospital data for improved patient-state forecasting, simulation, and clinical decision support.

## 2 Related Work

The integration of heterogeneous EHR data into machine learning models has been a significant research focus, particularly with the adoption of sequence models like transformers (Vaswani, 2017). Existing methods fall into two categories: *Outcome Prediction Models*, that target specific patient outcomes, such as mortality,

length-of-stay or diagnosis prediction, and *Patient Timeline Prediction Models*, which forecast full health trajectories but remain under-explored.

**Outcome Prediction Methods**   Most EHR models predict outcomes using structured data like medical codes. Models such as BEHRT (Li et al., 2020), Med-BERT (Rasmy et al., 2021), CEHR-BERT (Pang et al., 2021), TransformEHR (Yang et al., 2023) have enhanced accuracy with temporal embeddings, artificial time tokens and multi-task learning. EHRSHOT (Wornow et al., 2023) introduced a benchmark integrating structured demographics and various coded events. Some other works include additional numerical or categorical data (Li et al., 2022; Pellegrini et al., 2023; Lovón-Melgarejo et al., 2024). Another major challenge in EHR modeling is *Long-Context Integration*, due to EHR's time span and volume. Some outcome prediction models like Hi-BEHRT (Li et al., 2022), EhrMamba (Fallahpour et al., 2024), and CONTEXT (Wornow et al., 2024) extend token limits with hierarchical or sub-quadratic architectures, while models like REMed (Kim et al., 2024) and EMERGE (Zhu et al., 2024) propose retrieval-augmented approaches for selecting relevant features. UniHPF and GenHPF (Hur et al., 2022; 2023) show the effectiveness of combining structured and unstructured data into a unified text-based event representation with promising results on outcome classification tasks. Despite these advances, outcome prediction models typically target rather narrow downstream tasks and operate on a limited feature space, often focusing only on a fixed set of structured medical codes or few continuous signals, excluding a large set of available EHR data. They generally do not model or generate full patient trajectories.

**Patient Timeline Prediction Methods**   In contrast to the prevalent outcome prediction methods, patient timeline prediction seeks to model and predict the entire sequence of a patient's health events, offering a more holistic view of health trajectories. For instance, ESGPT (McDermott et al., 2023) provides a flexible library for transformer-based modeling of continuous-time event streams, including pre-processing utilities and example architectures. Recent approaches like CEHR-GPT (Pang et al., 2024) adapt GPT models for synthetic EHR generation, using structured tokens enriched with lab results, temporal and demographic information, but do not target patient-specific clinical forecasting. MOTOR (Steinberg et al., 2023) trains on timestamped sequences of medical events for time-to-event prediction, targeting individual outcomes like death or lab tests. Foresight (Kraljevic et al., 2024) predicts biomedical concepts from clinical notes, converting text into structured sequences using the SNOMED ontology (El-Sappagh et al., 2018). ETHOS (Renc et al., 2024) models structured EHR data from MIMIC-IV, generating tokenized Patient Health Timelines (PHTs). Yet, it omits unstructured notes and ICU chart events, key sources of patient information, and is limited to a 2,048-token input window. While these methods represent important progress toward trajectory modeling, they often operate on constrained input lengths, exclude unstructured modalities such as free-text notes and dense time-series signals such as ICU chart events and do not assess long-term simulation capabilities.

Our work shifts the focus to forecasting and evaluating on entire hospital pathways, moving beyond isolated outcome prediction towards a comprehensive understanding of patient development. Unlike prior methods, we comprehensively integrate all available structured and unstructured EHR data and leverage LLMs to understand this heterogeneous and noisy information in its native form. To address long-context challenges, we introduce a novel summary mechanism that enables efficient modeling over extended time horizons and complex patient histories.

## 3    Comprehensive Pathway-Level Modeling of Health Trajectories

We propose a novel approach to predict longitudinal patient health trajectories by integrating heterogeneous EHR data into a structured, language-based representation, as shown in Figure 2. Using an LLM backbone, we leverage its contextual understanding to model complex health data. To capture extended time horizons, we introduce a Masked Summarization Bottleneck that efficiently compresses longitudinal data. Additionally, we introduce two fine-tuning strategies for specializing EHR2Path to specific tasks. Our method enables comprehensive trajectory simulation, supporting holistic patient modeling.

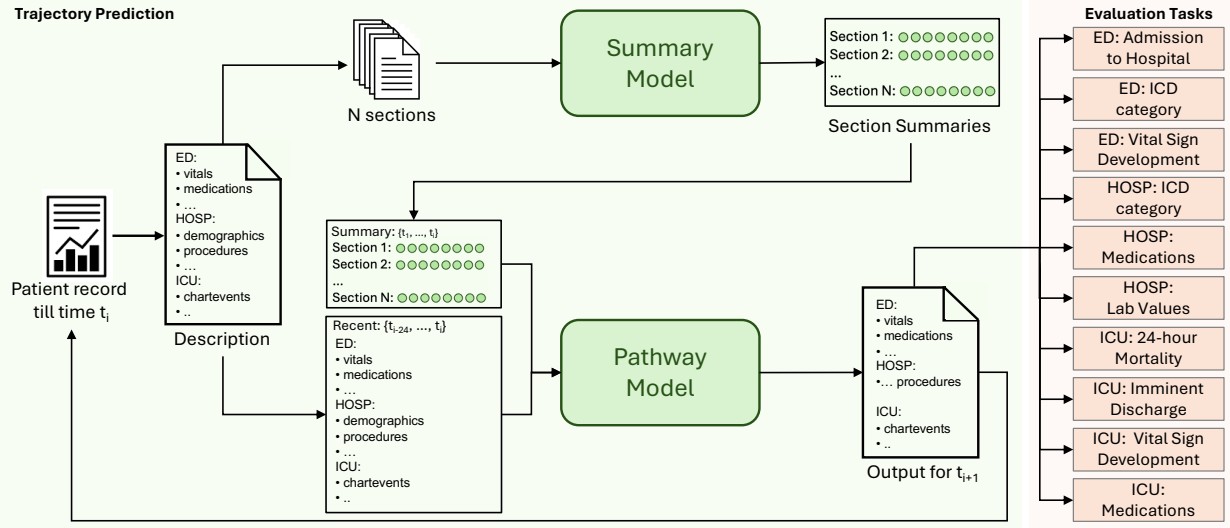

Figure 2: Overview of our proposed method. A patient record is structured into text from which a fixed time window is kept as text representation, while the full temporal context is summarized into an embedding-based summary by the summary model. The Pathway Model combines both representations to predict the next time-step. For iterative simulation, predictions update the patient record to simulate future trajectories until a termination condition is met.

## 3.1 Data Representation

EHR data is inherently heterogeneous, including static and temporal data, numerical and categorical features, and unstructured text from various clinical settings. We develop a unified textual representation of this data preserving its original form, avoiding heavy pre-processing to embrace real-world noise and incompleteness. This enhances robustness and supports diverse inputs without feature-specific adjustments, thereby improving scalability and real-world applicability. A central design choice is broad coverage of multimodal patient state rather than restricting the representation to a small hand-selected variable set. This increases representation size, especially for dense ICU stays with many simultaneously observed variables, but allows the model to preserve high-dimensional and informative patient state. Our method supports diverse data types and uses natural language for feature names and values, instead of medical codes, enabling semantic interpretation by an LLM. An exemplary data sample is shown in appendix A.7.

**Hierarchical Data Organization**  We denote the recorded data for patient $p$ as $D_{p,t} \mid t \in T_p$, where $t$ denotes the current hour of the stay and $T_p$ the total stay time of patient $p$. $D_{p,t}$ is hierarchically organized into three levels:
***Clinical Units***: At the highest level, $D_{p,t}$ is partitioned into clinical units $U = \{\text{ED}, \text{Hospital}, \text{ICU}\}$, representing the Emergency Department, Hospital, and Intensive Care Unit.
***Data Categories***: At each clinical unit $u \in U$, the data is subdivided into categories $C_u$ such as Demographics, Vital Signs, Prescriptions, Procedures, Radiology Reports, and Chart Events.
***Features***: Each category $c \in C_u$ consists of one or multiple features $F_{u,c} = \{f_1, f_2, \ldots, f_n\}$. We consider two types of features, categorized as *Events* or *Values*. *Events*, e.g. treatments or reports, do not record values and are directly included in the data representation. *Values* require value recording and may represent binary indicators *(e.g., "ST Segment Monitoring On: yes/no")*, continuous numerical values *(e.g., "Heart Rate: 82")* or categorical variables *(e.g., "Heart Rhythm: Sinus Rhythm")*, capturing the diversity of EHR data.

**Temporal Features**  Most information in a patient's EHR varies over time. Such temporal data is modeled as a sparse feature sequence $\mathbf{Z}_f = \{z_{f,t} \mid t \in \{1, 2, \ldots, T_p\}\}$, where each $z_{f,t}$ represents the recorded value or event at timestamp $t$, where $t$ specifies how many hours have passed since the value was recorded relative to the current time-point (e.g., *"Heart Rate: **1:** 80"* for the previous hour). Consecutive identical values

are merged into intervals (e.g., *"Heart Rate: 10–4: 82"*). Missing data is indicated by the absence of a corresponding entry in $\mathbf{Z}_f$ *(e.g., "Heart Rate: **5**:82, **1**:80").*

**State Attributes and Diagnosis Codes** We incorporate state attributes to mark significant changes in the patient's trajectory. These include events such as discharge from the ED, admission to and discharge from the ICU and Hospital, as well as death. At the final time step $T_p$ of a stay, ICD categories (Slee, 1978) corresponding to the patient's diagnosis are added.

**Input and Output Preparation** Model input and output are constructed by generating textual representations following the hierarchical structure defined above. The input includes all available data for a time window between $t - w$ up to the current time-point $t$, where $w$ denotes the input window size. To integrate longer time windows, we introduce a *Masked Summarization Bottleneck*, which condenses the entire patient history into one or multiple embedded representations for each category consisting of $N$ summary tokens each, allowing a window size of $t$. The output is formatted similarly but exclusively contains data recorded at the subsequent time-point $t + 1$. To enhance efficiency, the output is sparse, including only the features that were actively recorded at $t + 1$.

## 3.2 Patient Trajectory Prediction

Building upon the structured, language-based EHR representation outlined in section 3.1, we now detail our approach for predicting future patient states and clinical pathways.

### 3.2.1 Model Architecture and Training

The core component of our architecture is the *Pathway Model*, a transformer-based LLM tailored for patient trajectory prediction. It is trained to predict a sparse, structured textual output for the subsequent time-point $t + 1$, including all actively recorded features $z_{f,t+1}$ (e.g., lab values, procedures) and state transitions (e.g., discharge, transfer, or death), leveraging all available data from the patient record $D_{p,t}$. The training objective is to forecast all EHR elements recorded at $t + 1$, fostering a comprehensive understanding of the patient's clinical pathway. We propose and compare three model variants, each utilizing different inputs to the pathway model:

**EHR2Path-Text (E2P-T)**: Processes a text representation of static patient data and events within the most recent $w$ hours (e.g., $[t - w, t]$), capturing detailed recent observations in a structured format. Unless otherwise noted, *E2P-T* denotes the 24-hour text-only model, where $w = 24$; alternative text windows are written explicitly, e.g., *E2P-T-1h*.

**EHR2Path-Summ (E2P-S)**: Relies on summary embeddings of the entire patient history, generated via the *Masked Summarization Bottleneck* (section 3.2.2), efficiently integrating long-term context.

**EHR2Path-Summ+Text (E2P-S+T)**: Combines the text representation of recent events covering the 24-hour context used in *E2P-T* and the full history summary, balancing detailed short-term data with comprehensive longitudinal context.

Appendix A.8 provides detailed implementation and training details.

**Inference for Trajectory Simulation** To simulate full patient trajectories, the Pathway Model is applied iteratively at each time step $t$ until a stopping criterion is met, such as hospital discharge, death, or a predefined number of steps. At each iteration, the Pathway Model predicts the structured output for $t + 1$, which is reintegrated into the patient record, yielding $D_{p,t+1}$. The new patient record $D_{p,t+1}$ is processed according to the model variants to form a new input. This iterative procedure generates a synthetic sequence of future states $\{z_{f,t+1}, z_{f,t+2}, \dots\}$ for all relevant features $f$, effectively simulating the patient's clinical course.

### 3.2.2 Masked Summarization Bottleneck

For *E2P-S* and *E2P-S+T*, we propose a *Masked Summarization Bottleneck* that produces compact, task-relevant representations for each clinical unit $u$ and data category $c$ to handle extensive patient histories. Formally, let $\mathbf{X}_t = \{x_1, x_2, \dots, x_n\}$ be the $n$ token long description of a section in the patient's record at time $t$. We append $m$ *summary tokens* $\mathbf{S} = \{s_1, \dots, s_m\}$ and $o$ *output tokens* $\mathbf{Y} = \{y_1, \dots, y_o\}$ describing the values at $t + 1$ for the current section, forming the sequence $\mathbf{X}_t^{\text{full}} = \{\mathbf{X}_t, \mathbf{S}, \mathbf{Y}\}$. A custom attention mask $\mathbf{M}$

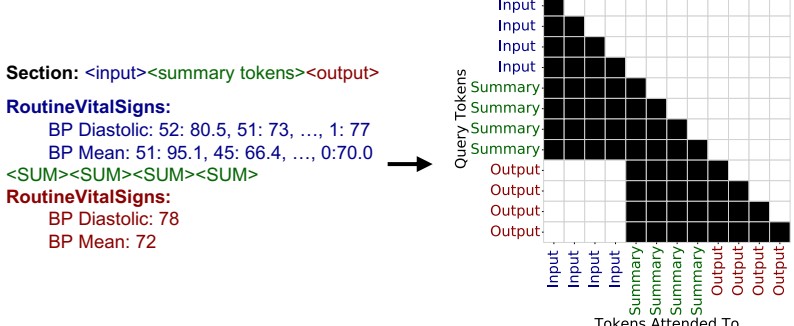

Figure 3: Masked Summarization Bottleneck. Input tokens encode past observations, while summary tokens (<SUM>) compress key information. A custom attention mask ensures outputs attend only to summaries, forcing the model to encode relevant patient data into a compact representation.

constrains how tokens $i$ can attend to tokens $j$:

$$
M_{ij} = \begin{cases} 1 & \text{if } j \leq i \text{ and } i \leq n + m, \\ 1 & \text{if } j \leq i \text{ and } j > n, \\ 0 & \text{otherwise.} \end{cases}
$$

As visualized in fig. 3, this ensures that output tokens $\{y_1, \ldots, y_o\}$ can only attend to the summary tokens $\{s_1, \ldots, s_m\}$ and preceding output tokens, creating a bottleneck. Following the information bottleneck principle (Tishby et al., 1999), our approach derives a compact representation $\mathbf{S}$ that maximizes mutual information $I(\mathbf{S}; \mathbf{z}_{t+1})$ with the next state $\mathbf{z}_{t+1}$ while constraining $I(\mathbf{S}; \mathbf{Z}_{1:t})$ with the patient history $\mathbf{Z}_{1:t}$ subject to the capacity $m$. Consequently, the Summarization Module is driven to compress the most relevant information to predict future states into the summary tokens, while discarding uninformative or redundant details. During inference, the summary tokens provide a lightweight embedding of the entire patient history for each data category $c$ in $D_{p,t}$. By handling each section separately, this allows us to in most cases include the entire patient history. At inference, the input tokens $\{x_1, \ldots, x_n\}$ for each category $c$ are combined with the summary tokens $\{s_1, \ldots, s_m\}$, and a single forward pass computes their hidden states $\mathbf{H} = \{\mathbf{h}_1, \ldots, \mathbf{h}_{n+m}\}$. The hidden states of the summary tokens, $\mathbf{H}^{\text{summary}} = \{\mathbf{h}_{n+1}, \ldots, \mathbf{h}_{n+m}\}$ are the final output of the Summary Model. This enables efficient summarization of all sections in $D_{p,t}$ without separate models, thus reducing computational overhead. Unlike traditional auto-encoder approaches that reconstruct every detail of the input, we optimize summary tokens for forecasting the patient's next state. This task-focused approach avoids the inefficiencies associated with generic reconstructions and produces more effective representations for patient pathway prediction.

### 3.2.3 Length-of-Stay Indicator

Converging to final states like discharge requires accurate state predictions across multiple forecast iterations, which is challenging because such events are relatively rare compared to typical value changes, often causing the model to inadvertently prolong patient stays without termination. We tackle this by appending a Length-of-Stay (LOS) indicator, a simple countdown of remaining hours for each clinical unit $u$. To increase robustness, and allow the model to work without using the ground-truth LOS token during inference, we perform two augmentations. In half of the samples, we drop the LOS token entirely from the input, and in the other half, we inject noise (+-20%) into the LOS token, encouraging the model to re-estimate the remaining LOS at each time step rather than rigidly decrementing it. Crucially, at inference time, we never include the ground truth LOS token in the input, instead the first step is prompted without LOS token, while in later steps the predicted LOS tokens of prior steps are included.

### 3.2.4 Fine-tuning for Outcome Prediction

While our primary focus is on trajectory simulation, the model can also serve as a foundation for downstream prediction tasks through targeted fine-tuning. We explore two complementary fine-tuning strategies: *Specialized Pathway Fine-Tuning*, fine-tuning on curated pathway data in the original format (e.g., only Emergency Department cases), to specialize for specific outcomes while retaining insights into intermediate events; and *Outcome-Oriented Fine-Tuning*, simplifying the task to a single-step outcome prediction to maximize predictive performance.

## 4 Experimental Setup

### 4.1 Dataset and Pre-processing

We use the MIMIC-IV database (Johnson et al., 2023b), containing de-identified, real-world EHRs for approximately 300,000 patients, as it is one of the largest and the most comprehensive EHR databases to date, offering granular and heterogeneous patient records, including coded, numerical and free-text features from different clinical units. We incorporate all clinically relevant tables, encompassing 22 tables and 20 ICU Chart Event categories (listed in appendix A.1), including admissions, lab results, medications, procedures, vital signs, and more, spanning the Emergency Department (ED), hospital wards, and Intensive Care Unit (ICU). Further, its inherent noisiness and sparsity strengthen its value as a realistic benchmark. To retain this real-world complexity, we avoid applying exclusion criteria and preserve missing or incorrectly populated fields. Following previous work (Wang et al., 2020; McDermott et al., 2021), we aggregate values hourly using average for numerical values, and most frequent for categorical ones, managing computational feasibility by balancing temporal resolution and recording frequency. Average context lengths are 380 tokens per hour, 1,880 for 24 hours, or 11,361 for the full history. Data is split at the patient level: 95% for training, 2.5% each for validation and test sets. Training samples consist of EHR data $D_{p,t}$ up to time $t$, with $t+1$ as the label. In practice, since patient stays often span hundreds of hours, using all hourly slices is computationally infeasible. Instead, we sample time points from all patients using weighted sampling, where time points are weighted based on the rarity of clinical events in the output, with rarer events receiving higher log-scaled weights. Additionally, we oversample critical transitions and key events such as admissions, discharges, and deaths to ensure adequate representation in training. Overall, we collect one million weighted samples for training, while fixed sets of 5000 val/test time points each are sampled without weighting, preserving the original distribution.

### 4.2 Longitudinal Simulation Tasks

We define nine clinically relevant evaluation tasks across ED, Hospital, and ICU, focusing on outcome and development prediction, partially inspired by McDermott et al. (2021). For each task we perform 500 simulations to assess iterative trajectory prediction. ED tasks use only ED data, while Hospital and ICU tasks incorporate all prior data. Detailed task descriptions can be found in appendix A.9.
**Outcome Prediction Tasks**: ED Admission *(will ED patient be hospitalized)*, ED/Hospital Discharge Diagnosis (multi-label prediction over 18 ICD categories), Imminent Mortality (24 hours), Imminent Discharge (3 days).
**Development Prediction Tasks**: 24-hour forecasts for ED Vital Signs, Hospital Medications, Hospital Lab Values, ICU Vital Signs, ICU Inputs.

### 4.3 Baselines

For comparison, we select trajectory and outcome prediction methods trained on the MIMIC-IV dataset with publicly available code, enabling reproduction on our data splits and tasks: **ETHOS** (Renc et al., 2024), a transformer-based model for tokenized health timeline simulation, trained on next-token prediction. We compare against ETHOS on next time-step prediction and all applicable trajectory simulation tasks. **MEME** (Lee et al., 2024), an LLM-based approach for outcome classification on textualized Emergency Department data, fine-tuned to predict outcomes from the full textual summary of ED records. **REMed** (Kim

et al., 2024), a transformer-based long context model for ICU outcome prediction that scores EHR events using an MLP and feeds the most important ones to a transformer for classification. We compare fine-tuned EHR2Path variants with MEME and REMed on applicable outcome prediction tasks.

### 4.4 Evaluation Metrics

We assess performance on next-timestep prediction and longitudinal simulation using tailored metrics:

**Next-Timestep Prediction Metrics.**
***Event Prediction:*** To accurately represent false positive and negative rates, we use micro- and macro-averaged F1 scores for predicting which events or features will be present in the next hour.
***Event+Value Prediction:*** Measures overall correctness by requiring both correct event timing and value prediction. Numerical values use a modified MAE (correct-time predictions scored by MAE; incorrect/missing predictions receive maximum error). Numerical values are normalized with 1st–99th percentile clipping and min-max scaling to [0,1]. Categorical values use modified accuracy with missing predictions counted as incorrect. Macro metrics are averaged at feature level to provide an unbiased assessment across diverse features. Additionally we report the average and maximum captured context and required input tokens, measured as tokens in the tokenized sequence. Context refers to the token count of the text representation of the input window, and input tokens to the final model input after potential summarization.

**Longitudinal Simulation Metrics.** For extended trajectory forecasting, we use task-specific metrics. Binary classification tasks (e.g., mortality prediction) are evaluated using accuracy, sensitivity and specificity on balanced test sets, while regression-based tasks (e.g., vital sign forecasting) are assessed using event-based F1, MAE on min-max normalized values, and accuracy. For binary outcome tasks evaluated with accuracy, we use a balanced test set for sample efficiency. Value predictions are only evaluated if a matching time-point exists, with a $\pm 1$ hour tolerance to account for EHR recording variability.

## 5 Results and Discussion

We evaluate EHR2Path on MIMIC-IV, jointly modeling emergency department (ED), ward, and ICU pathways as a single longitudinal trajectory. To provide a high-level overview of our evaluation, fig. 4 summarizes both task coverage and performance across the main simulation and outcome tasks. EHR2Path is evaluated on a substantially broader set of in-hospital pathway tasks than prior baselines, while remaining competitive or superior on directly comparable settings. A detailed analysis of these task groups follows in later sections.

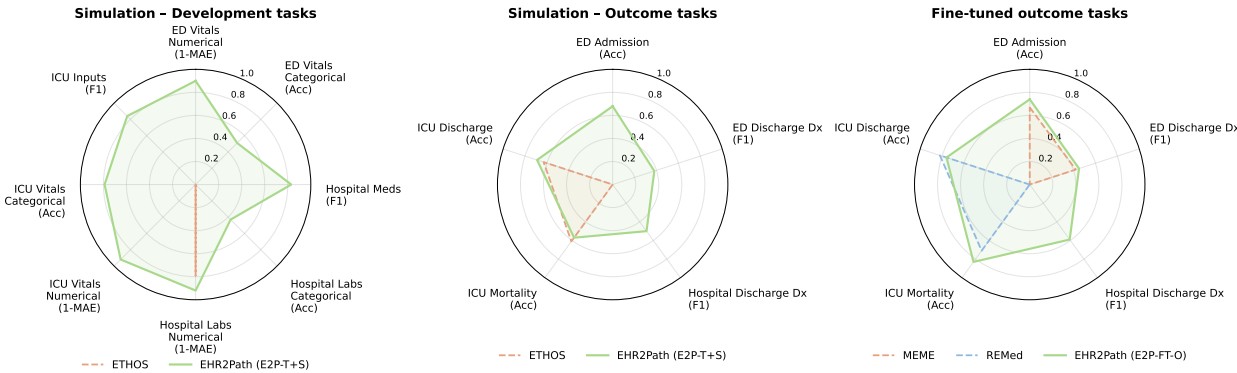

Figure 4: Overview of the evaluation across simulation and outcome tasks, summarizing task coverage and performance of EHR2Path relative to prior baselines on the main task groups. We report the primary metric per task, with numerical tasks reporting $1-\mathrm{MAE}$. Unsupported tasks are plotted as zero to visualize task coverage.

Table 1: Next Time-Step Prediction: We report event F1, event-value scores for numerical and categorical predictions and the avg./max. context and input tokens required to capture this context. For ETHOS, results are reported (i) in the full pathway-modeling setting (upper part), where outputs ETHOS can not forecast are reflected as negative instances and (ii) in an ETHOS-compatible matched-output comparison against E2P-S+T. Unless otherwise specified, E2P-T denotes the 24-hour text-only model. 95% confidence intervals are given in brackets.

| | F1 macro/micro | Numerical ↓ macro/micro | Categorical macro/micro | Avg./Max. context | Avg./Max. input |
|---|---|---|---|---|---|
| Statistic | 0.02 / 0.02 | 0.99 / 0.97 | 0.03 / 0.16 | n/a | n/a |
| ETHOS | 0.003 (0.00,0.008) / 0.03 (0.03,0.03) | 1.00 (0.99,1.00) / 0.97 (0.97,0.97) | 0.04 (0.00,0.12) / 0.31 (0.31,0.32) | 1217/2048 | 1217/2048 |
| E2P-T-1h | 0.42 (0.30,0.53) / 0.64 (0.64,0.64) | 0.67 (0.52,0.80) / 0.89 (0.88,0.89) | 0.30 (0.16,0.44) / 0.52 (0.51,0.53) | 380/3142 | 380/3142 |
| E2P-T | 0.47 (0.36,0.57) / **0.78** (0.78,0.78) | **0.64** (0.51,0.77) / 0.74 (0.74,0.75) | **0.33** (0.21,0.47) / **0.57** (0.56,0.57) | 1365/4250 | 1365/4250 |
| E2P-S | **0.48** (0.36,0.59) / 0.76 (0.76,0.77) | 0.66 (0.55,0.78) / **0.72** (0.71,0.72) | **0.33** (0.21,0.46) / 0.56 (0.55,0.56) | 9823 / 86621 | 220 / 1494 |
| E2P-S+T | 0.43 (0.33,0.54) / **0.78** (0.78,0.78) | 0.68 (0.58,0.79) / 0.73 (0.72,0.73) | **0.33** (0.21,0.46) / 0.55 (0.55,0.56) | 9823/86621 | 1414/4313 |
| Restricted to data supported by ETHOS: | | | | | |
| ETHOS | 0.04 (0.00,0.08) / 0.08 (0.08,0.09) | 0.96 (0.96,0.96) / 0.96 (0.96,0.96) | 0.86 (0.86,0.86) / 0.85 (0.84,0.86) | 1217 / 2048 | 1217 / 2048 |
| E2P-S+T | **0.12** (0.00,0.24) / **0.24** (0.23,0.25) | **0.87** (0.87,0.87) / **0.87** (0.87,0.88) | **0.99** (0.99,0.99) / **0.93** (0.93,0.94) | 9823/86621 | 1414/4313 |

## 5.1 Next Timestep Prediction Results

We compare against a statistical baseline (sampling events by occurrence frequencies and using mean/majority values) and ETHOS (Renc et al., 2024), a transformer-based patient timeline simulator. The poor performance of the statistical baseline highlights the difficulty of the task. When evaluated on the full MIMIC-IV feature space, ETHOS underperforms substantially across all metrics, largely because it cannot predict many EHR event types, which is reflected in a very low F1 score. Restricting evaluation to the subset of data supported by ETHOS improves its performance slightly, but E2P-S+T still performs better while predicting a substantially broader range of variables. Among our models, E2P-S achieves the highest macro F1 and micro numerical score, highlighting the effectiveness of the Masked Summarization Bottleneck in extracting predictive information from the full patient history. E2P-T (24h) achieves strong micro F1 and outperforms E2P-T-1h, indicating the benefit of access to a longer recent context. E2P-S+T performs comparably to E2P-T (24h), but does not surpass E2P-S overall. In this context, E2P-T (1h) and E2P-T (24h) serve as truncation-style baselines that allocate the available budget entirely to recent raw text, while E2P-S tests whether the learned summary representation retains useful predictive information on its own. The fact that E2P-S remains clearly predictive supports that the summary tokens capture meaningful signal. Importantly, table 1 also reports both the amount of true historical context available to each model and the number of tokens actually passed to the model. Part of this token cost arises from prioritizing broad coverage of heterogeneous patient state, so even short windows can be token-heavy when they contain many simultaneously observed variables, especially in ICU settings. All pathway variants use the same input budget, thus, in the mixed variant, summary tokens use a part of that budget and replace some of the raw-text input rather than increasing total prompt length. Under this fixed-budget setting, while text-only variants reason in detail over recent windows, summary-based variants compress substantially longer trajectories into compact forecasting-oriented representations. Concretely, E2P-S and E2P-S+T can incorporate up to 20× more historical context on MIMIC-IV. These results therefore support the bottleneck as a mechanism for fixed-budget context efficiency, improving the amount of longitudinal information represented without systematically increasing input length.

Table 2: Simulation-based Results.

| | ETHOS | E2P-T | E2P-S | E2P-S+T |
|---|---|---|---|---|
| *ED Vital Sign Development* | | | | |
| Event F1 | n/a | **0.61** (0.58,0.64) | 0.53 (0.49,0.56) | 0.57 (0.54,0.60) |
| Value MAE ↓ | n/a | **0.10** (0.10,0.10) | 0.12 (0.12,0.13) | **0.10** (0.10,0.11) |
| Value Acc. | n/a | **0.54** (0.48,0.61) | **0.54** (0.46,0.61) | 0.51 (0.44,0.57) |
| *ED Admission Prediction* | | | | |
| Acc. | n/a | 0.67 (0.63,0.71) | 0.63 (0.59,0.67) | **0.68** (0.64,0.72) |
| Sens. | n/a | 0.58 (0.52,0.64) | 0.48 (0.42,0.54) | **0.63** (0.57,0.69) |
| Spec. | n/a | **0.80** (0.76,0.85) | 0.78 (0.73,0.83) | 0.73 (0.68,0.78) |
| *ED Discharge Diagnosis* | | | | |
| F1 | n/a | **0.41** (0.37,0.45) | 0.28 (0.24,0.32) | 0.38 (0.33,0.42) |
| *Hospital Medication Development* | | | | |
| Event F1 | n/a | 0.83 (0.80,0.85) | 0.82 (0.79,0.85) | **0.83** (0.80,0.85) |
| *Hospital Lab Value Development*[†] | | | | |
| Event F1 | 0.05 (0.03,0.07) | 0.20 (0.16,0.25) | 0.17 (0.13,0.20) | **0.23** (0.19,0.27) |
| Value MAE ↓ | 0.21 (0.19,0.23) | 0.09 (0.08,0.09) | 0.11 (0.11,0.12) | **0.08** (0.08,0.09) |
| Value Acc. | n/a | 0.32 (0.00,1.00) | 0.36 (0.00,1.00) | **0.43** (0.13,0.81) |
| *Hospital Discharge Diagnosis* | | | | |
| F1 | n/a | 0.49 (0.47,0.51) | 0.47 (0.45,0.49) | **0.50** (0.48,0.52) |
| *ICU Vital Sign Development* | | | | |
| Event F1 | n/a | 0.70 (0.67,0.72) | **0.75** (0.73,0.78) | 0.71 (0.69,0.73) |
| Value MAE ↓ | n/a | 0.09 (0.09,0.09) | 0.10 (0.09,0.10) | **0.08** (0.08,0.08) |
| Value Acc. | n/a | 0.69 (0.66,0.72) | **0.80** (0.77,0.82) | 0.79 (0.77,0.81) |
| *ICU Input Development* | | | | |
| Event F1 | n/a | 0.79 (0.76,0.81) | **0.85** (0.83,0.87) | 0.84 (0.82,0.86) |
| *ICU Imminent Mortality* | | | | |
| Acc. | **0.61** (0.57,0.65) | 0.53 (0.49,0.58) | 0.50 (0.46,0.54) | 0.57 (0.53,0.62) |
| Sens. | **0.23** (0.18,0.29) | 0.08 (0.05,0.12) | 0.00 (0.00,0.00) | 0.16 (0.12,0.21) |
| Spec. | 0.99 (0.97,1.00) | **1.00** (1.00,1.00) | **1.00** (1.00,1.00) | 0.98 (0.97,1.00) |
| *ICU Imminent Discharge* | | | | |
| Acc. | *0.63* (0.51,0.74) | 0.63 (0.59,0.67) | 0.57 (0.53,0.61) | **0.69** (0.65,0.73) |
| Sens. | 0.71 (0.54,0.87) | 0.46 (0.40,0.52) | 0.27 (0.21,0.33) | **0.82** (0.77,0.86) |
| Spec. | 0.56 (0.38,0.72) | 0.70 (0.64,0.76) | **0.88** (0.83,0.92) | 0.56 (0.50,0.62) |

Table 3: Fine-Tuning for Outcome Tasks.

| | MEME | REMed | E2P-FT-P | E2P-FT-O |
|---|---|---|---|---|
| *ED Admission Prediction* | | | | |
| Acc. | 0.67 (0.63,0.71) | n/a | 0.73 (0.69,0.77) | **0.74** (0.70,0.77) |
| Sens. | 0.59 (0.53,0.65) | n/a | 0.72 (0.67,0.78) | **0.74** (0.68,0.79) |
| Spec. | **0.76** (0.70,0.81) | n/a | 0.73 (0.68,0.78) | 0.74 (0.68,0.79) |
| *ED Discharge Diagnosis* | | | | |
| F1 | 0.42 (0.37,0.48) | n/a | $(0.45)^*$ (0.41,0.48) | **0.45** (0.41,0.48) |
| *Hospital Discharge Diagnosis* | | | | |
| F1 | n/a | n/a | $(0.59)^*$ (0.56,0.61) | 0.59 (0.56,0.61) |
| *ICU Imminent Mortality* | | | | |
| Acc. | n/a | 0.71 (0.60,0.80) | 0.77 (0.74,0.81) | **0.83** (0.80,0.86) |
| Sens. | n/a | 0.41 (0.25,0.57) | 0.64 (0.59,0.70) | **0.72** (0.67,0.78) |
| Spec. | n/a | **1.00** (1.00,1.00) | 0.90 (0.87,0.94) | 0.94 (0.91,0.97) |
| *ICU Imminent Discharge* | | | | |
| Acc. | n/a | **0.82** (0.78, 0.85) | 0.69 (0.65,0.73) | 0.76 (0.72,0.79) |
| Sens. | n/a | **0.85** (0.80,0.90) | 0.80 (0.75,0.85) | 0.78 (0.72,0.83) |
| Spec. | n/a | **0.81** (0.77,0.85) | 0.58 (0.52,0.64) | 0.74 (0.69,0.80) |

*Table 2:*
[†] *Restricting hospital-lab evaluation to the ETHOS-supported lab vocabulary (150 labs) changes results only minimally (<0.01) and leaves normalized MAE unchanged for both ETHOS and E2P-S+T.*
*Table 3:*
*FT-P: Specialized Pathway Fine-Tuning.*
*FT-O: Outcome-Oriented Fine-Tuning.*
*n/a: Indicates the model is not applicable to the task as it cannot integrate or predict the relevant data.*
*95% confidence intervals are given in brackets.*
*[*]Diagnosis prediction is inherently a one-step task making pathway- and outcome-fine-tuning identical.*

## 5.2 Patient Trajectory Simulation Results

E2P-T, E2P-S, and E2P-S+T all perform strongly on simulation tasks (table 2), with distinct strengths: E2P-S excels in long-term, dense-context tasks such as ICU Vital Sign and Input Development, while E2P-T outperforms in immediate state predictions such as ED Admission or Diagnosis tasks. The combined model, E2P-S+T consistently ranks first or second, often achieving the best results. This stability makes it a robust

compromise, balancing text- and summary-based strengths for strong performance across diverse simulations. Taken together, the results show strong simulation performance across different clinical units, including ED, hospital/ward, and ICU settings. For the binary simulation tasks, we additionally report sensitivity and specificity to better characterize rare and clinically important outcomes. The text-only and summary-only variants, E2P-T and E2P-S, often emphasize either sensitivity or specificity, whereas the combined model, E2P-S+T, is generally more balanced, aligning with its strongest overall performance across the binary simulation tasks. A qualitative example of a vital sign simulation of E2P-S+T is shown in appendix A.5. Within the subset of shared tasks, our models remain competitive with ETHOS while enabling a substantially broader range of simulations through more comprehensive data integration. Because broader multimodal input support is part of the pathway-modeling contribution of EHR2Path, these results reflect both the effect of increased supported input space and differences in modeling approach. Overall, these results highlight our strength in temporal modeling and simulation of patient pathways.

We further analyze robustness to increasing prediction horizon in fig. 5. For key development tasks, we gradually increase the horizon in four-hour steps and evaluate event F1 and value MAE at each horizon. As expected, performance decreases with longer horizons, with a more pronounced drop for event occurrence (event F1) than for numerical values. Importantly, the decline is gradual rather than catastrophic, and performance remains at a stable level up to a 24-hour horizon across ED, ward, and ICU.

Lastly, we assess rollout realism with an automated clinical plausibility analysis, reported in Appendix A.2. Across all development prediction tasks, generated `E2P-T+S` rollouts show low empirical violation rates of absolute value range, one-step value jumps, blood-pressure consistency, and medication state change frequency, with behavior broadly comparable to matched held-out windows.

### 5.3 EHR2Path as a Foundation Model for Outcome Prediction

While EHR2Path is designed primarily as a pathway simulator, many clinical applications ultimately require concise outcome predictions: whether a patient will be admitted, what their discharge diagnosis will be, or whether they are at imminent risk of deterioration or discharge from the ICU. We therefore test whether EHR2Path can serve as a general-purpose foundation model for such outcome tasks via task-specific fine-tuning. We report results for Specialized Pathway Fine-Tuning (E2P-FT-P) and Outcome-Oriented Fine-Tuning (E2P-FT-O), and the specialized outcome prediction models MEME (Lee et al., 2024) and REMed (Kim et al., 2024) in table 3. Both fine-tuning strategies demonstrate the model's ability to effectively specialize for specific outcomes, with the outcome fine-tuning outperforming pathway fine-tuning at the cost of lacking intermediate outcomes. Compared to the baselines, EHR2Path-FT-O outperforms in three of four tasks. These findings underscore the versatility of our approach beyond trajectory prediction, highlighting its potential as a robust foundation for a wide range of EHR-based prediction tasks. Considering sensitivity and specificity, our strongest outcome model, E2P-FT-O, shows a favorable trade-off between sensitivity and specificity across tasks. It achieves the strongest ED admission performance and substantially improves ICU mortality detection over pathway fine-tuning. Compared to REMed, which attains much lower sensitivity but perfect specificity, E2P-FT-O provides a more balanced mortality detection trade-off while retaining high specificity. For ICU imminent discharge, REMed remains strongest overall, though E2P-FT-O has competitive results.

### 5.4 Additional Ablation Results

To assess the impact of key components, we conduct ablation studies on the *Length-of-Stay (LOS) Indicator* and the *Masked Summarization Bottleneck* size. We assess the LOS Indicator's impact on tasks requiring convergence to final states, *Hospital Discharge Diagnosis* and *ICU Imminent Discharge*. Failure to converge within realistic, dataset-derived step limits is counted as an incorrect prediction. As shown in table 5, removing the LOS Indicator significantly hinders convergence to a final state, reducing predictive accuracy.

For ICU Imminent Discharge, termination behavior is directly relevant because the rollout-derived label depends on whether ICU release is generated within the 72-hour horizon. We therefore decompose the test-set predictions into termination-outcome categories in table 4. The majority of rollouts fall into the two correct termination categories, while incorrect termination can be separated into premature predicted

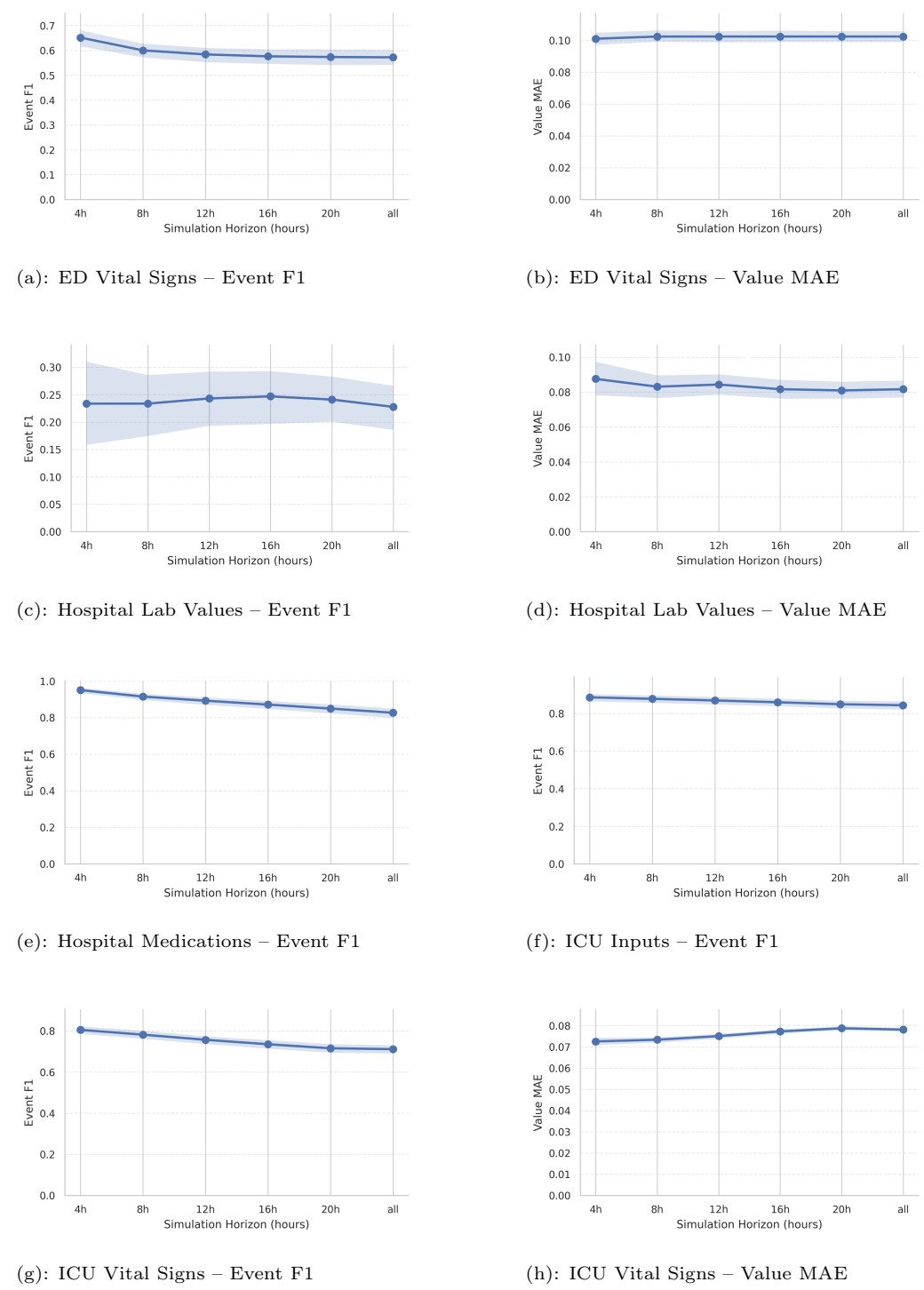

(a): ED Vital Signs – Event F1

(b): ED Vital Signs – Value MAE

(c): Hospital Lab Values – Event F1

(d): Hospital Lab Values – Value MAE

(e): Hospital Medications – Event F1

(f): ICU Inputs – Event F1

(g): ICU Vital Signs – Event F1

(h): ICU Vital Signs – Value MAE

Figure 5: Development of event detection (F1) and value prediction (MAE) across simulation horizons for ED, hospital, and ICU development tasks. The shaded areas indicate 95% confidence intervals.

discharge and missed/delayed discharge, with premature discharge being the more common error mode. The 21.8% premature predicted discharge rate remains a limitation of the current LOS-guided rollout procedure, indicating that the model can sometimes generate ICU release too early relative to the 72-hour decision window. Such errors can happen when the initial LOS indicator already under- or overestimates the remaining

Table 4: Termination-outcome analysis for ICU Imminent Discharge.

| Category | Count | Percentage |
|---|---|---|
| Correct imminent discharge | 204 | 40.8% |
| Correct non-imminent discharge | 141 | 28.2% |
| Premature predicted discharge | 109 | 21.8% |
| Missed/delayed discharge | 46 | 9.2% |

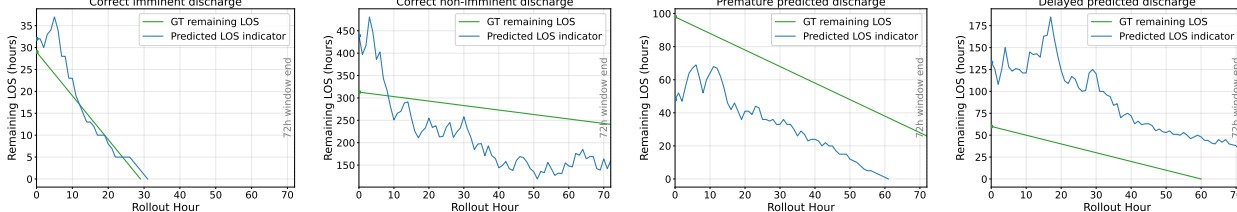

Figure 6: Qualitative LOS-indicator trajectories for the ICU Imminent Discharge task. These examples show how both the initial LOS estimate and the re-estimation during rollout can affect when ICU discharge is predicted.

stay, or when it decreases too quickly, stalls, or is re-estimated incorrectly during rollout. Figure 6 illustrates qualitative LOS trajectories across the four categories. Since the LOS indicator is regenerated at each rollout step, both its initial estimate and subsequent trajectory can affect whether a release state is generated before the 72-hour horizon. Overall, the trajectories often remain in an appropriate LOS range even when the exact remaining stay length is difficult to predict, such as in stays where discharge is still far in the future.

Further, we examine performance across various Masked Summarization Bottleneck sizes, trained for a limited number of steps. While longer contexts add detail, their quadratic cost limits scalability. Table 6 shows that, while larger bottleneck sizes slightly reduce validation loss, the most substantial improvement occurs from 4 to 8 tokens. Based on this, we select a bottleneck size of 8, balancing performance and efficiency.

### 5.5 Limitations and Future Work

An important consideration in our task formulation is that EHR2Path predicts the next recorded EHR state, and therefore the learned signal combines patient-state evolution with hospital measurement and documentation patterns. This is partly inherent to the task. Whether a lab test is ordered, a medication is administered, or a vital sign is charted is itself informative about clinical workflow and the tests or interventions considered necessary at a given time point. At the same time, this means the model should not be interpreted as recovering a fully observed physiological trajectory independent of documentation practices, as sparse outputs indicate what is expected to be recorded next, not necessarily all clinically true but unmeasured parts of the patient state.

Table 5: Effect of LOS Indicator.

| | | no LOS | with LOS |
|---|---|---|---|
| *Hospital Discharge Diagnosis* | F1 | 0.32 | **0.50** |
| | converged | 51% | 100% |
| *ICU Imminent Discharge* | Acc. | 0.65 | **0.69** |
| | converged | 29.6% | 68.5% |

Table 6: Effect of Bottleneck Size.

| Size | Val Loss | Avg./Max. Tokens |
|---|---|---|
| 1 | 0.27 | 11 / 91 |
| 4 | 0.27 | 42 / 364 |
| 8 | 0.21 | 86 / 728 |
| 16 | 0.20 | 172 / 1456 |
| 32 | 0.18 | 344 / 2912 |
| 64 | 0.18 | 688 / 5842 |

While the EHR2Path models establish a strong foundation for patient trajectory modeling, the focus on a single healthcare system may limit the generalizability of results to other healthcare settings, with differing demographics and site-specific treatment strategies, which could be addressed by collecting and using more comprehensive EHR data from a diverse set of countries and hospitals. Moreover, while the evaluated tasks demonstrate the feasibility of patient-pathway forecasting from structured EHR, they do not by themselves establish clear clinical utility for real-world decision-making. An important next step would be clinical validation. Any broader clinical use would require prospective validation to assess robustness, usefulness, and safety in practice. For adaptation to real hospital settings, an institution would need to validate models of this kind on its own data and intended use case, including subgroup performance and robustness under local documentation and care practices. Deployers would also be responsible for privacy-preserving data handling, appropriate governance and access controls, monitoring for performance drift after deployment, and ensuring the model is used with clinician oversight. Given the compact backbone used here, institution-hosted deployment could be used for privacy-sensitive settings. Finally, expanding the approach to include lifetime health data for long-term trajectory modeling is an exciting future direction.

## 6 Conclusion

In this work, we introduced a novel approach to patient pathway modeling, proposing a structured text-based representation of heterogeneous EHR data and a Masked Summarization Bottleneck to handle extended data categories and temporal contexts efficiently. By integrating diverse data modalities, including numerical, categorical, and free-text information, our model enables accurate next-step predictions and robust simulation of patient trajectories over extended time horizons. Our experiments demonstrate that this approach enables pathway-level modeling of long, heterogeneous in-hospital patient trajectories from multimodal EHR data on MIMIC-IV. By evaluating both next-step and long-term simulations, we show the feasibility of patient-state forecasting across a broad range of development and patient outcome prediction tasks. These results provide a basis for further research into comprehensive, task-independent EHR modeling.

### Broader Impact Statement

This work studies retrospective patient-state forecasting and trajectory simulation from multimodal in-hospital EHR data. Such systems may in the future support more personalized and predictive patient care. However, our results do not by themselves establish clinical utility, safety, or readiness for real-world decision support, as this would require use-case-specific clinical validation. Inappropriate use could create over-trust in inaccurate forecasts or simulated trajectories. Because these models rely on sensitive patient data, any real-world use would require strong privacy-preserving data handling, appropriate governance, and clinician oversight.

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

# A  Appendix

## A.1  Details on MIMIC-IV data

The MIMIC-IV Johnson et al. (2023b) dataset can be obtained via PhysioNet Goldberger et al. (2000); Johnson et al. (2023a) after performing the necessary credential process and CITI Data or Specimens Only Research training under the PhysioNet Credentialed Health Data License 1.5.0. We use version 2.2. of the dataset provided at `https://www.physionet.org/content/mimiciv/2.2/`, together with the corresponding versions of the MIMIC-IV-Note `https://www.physionet.org/content/mimic-iv-note/2.2/` and the MIMIC-IV-ED dataset `https://www.physionet.org/content/mimic-iv-ed/2.2/`) (Johnson et al. (2021)). In table 7, we list all tables from the MIMIC-IV (Johnson et al. (2023b)) dataset we include into our patient representation as well as the data categories included in the ICU chartevents table:

Table 7: Used tables and data categories from the MIMIC-IV dataset.

| Table name | Description |
|---|---|
| **Emergency department** | |
| EDSTAYS | Information about patient admissions to the ED, including in and out times, as well as patient demographics. |
| MEDRECON | Current medications a patient is taking at admission. |
| TRIAGE | Information about the initial triage at arrival in the ED, such as admission vital signs and the chief complaint. |
| PYXIS | Dispensed medications during the ED stay. |
| VITALSIGN | Routine vital signs taken every 1-4 hours in the ED. |
| DIAGNOSIS | All billed diagnoses for a patient. |
| **Hospital** | |
| OMR | Various measurements recorded outside the hospital. |
| ADMISSIONS | Information about hospital patient admissions, including partial demographics and admission and discharge times. |
| PATIENTS | Patients' age, gender and time of death. |
| SERVICES | The hospital service which cared for the patient in the hospital stay. |
| TRANSFERS | Information about transfers between different hospital unit. Can be different from the current service provider / care taker. |
| PRESCRIPTIONS | Prescribed medications during the hospital stay. |
| PROCEDURES | Performed procedures during the hospital stay. |
| LABEVENTS | Various laboratory measurements and test results. |
| MICROBIOLOGYEVENTS | Results of microbiology cultures. |
| DIAGNOSIS | Billed ICD-9/ICD-10 diagnoses for hospitalizations. |
| **ICU** | |
| ICUSTAYS | Information about ICU admissions, such as in and out time. |
| INPUTEVENTS | Information about continuous infusions or intermittent administrations in the ICU. |
| OUTPUTEVENTS | Information regarding patient outputs including urine, drainage, etc. |
| PROCEDUREEVENTS | Performed procedures during the ICU stay, such as ventilation or imaging exams. |
| CHARTEVENTS | Any charted items during the ICU Stay. We separate this data further into 20 categories: skin incisions, routine vital signs, skin impairment, cardiovascular, gi gu, toxicology, iabp, hemodynamics, respiratory, md progress note, adm history fhpa, dialysis, pain sedation, cardiovascular (pulses), pulmonary, cardiovascular (pacer data), nicom, alarms, skin assessment, neurological |

## A.2 Clinical Plausibility Analysis

To audit clinical plausibility, we evaluate whether generated rollout windows from `E2P-T+S` remain clinically and temporally consistent under simple empirical checks. We compared the test set generations against matched ground-truth windows on all development prediction tasks: ED vital signs, ICU vital signs, ICU inputs, hospital medications, and hospital labs.

For ED vital signs, ICU vital signs, and hospital labs, we consider two kinds of consistency. First, we measure *range-support violations*, where a value is flagged if it falls outside the empirical 1st - 99th percentile interval for that variable, using MIMIC-derived percentiles. Second, we measure *one-step jump violations*, where for each variable, we compute the empirical 99th percentile of absolute one-step changes on real windows and flag changes exceeding this threshold. These percentile-based tests indicate whether generated values in multi-step simulations remain plausible and within the data support. As an exemplary test for physiologic consistency, we do a blood pressure audit, testing whether systolic $\geq$ diastolic pressure, and if mean arterial pressure is between diastolic and systolic pressure when all three are present. Lastly, we evaluate if binary treatment trajectories exhibit implausibly noisy on/off switching. For ICU inputs and hospital medications, we count medication state changes between consecutive time points and flag rollout windows whose total change count exceeds 99th percentile for that task, indicating unusually many medication changes.

Across the audited tasks, the observed violation rates were generally low and broadly comparable to the matched real windows. As shown in table 8, generated ED and ICU vital-sign rollouts showed low rates of both range and change violations, and no generated blood-pressure consistency violations are observed. Hospital lab rollouts likewise exhibit low range and jump violation rates. For medication tasks, state change outlier rates are low for both generated and real windows. Taken together, these findings suggest that the generated trajectories remain clinically plausible and temporally consistent under the implemented empirical checks and behave similar to the real data.

Table 8: Automated clinical plausibility checks for `E2P-T+S`. We report exact counts and rates for each implemented check, comparing generated rollout windows against matched held-out windows.

| Task | Check | Generated | Real |
|---|---|---|---|
| ED vital signs | Range-support violations | 0.47% | 1.07% |
| ED vital signs | One-step jump violations | 0.39% | 1.08% |
| ED vital signs | Blood-pressure consistency violations | 0.00% | 0.00% |
| ICU vital signs | Range-support violations | 0.28% | 1.69% |
| ICU vital signs | One-step jump violations | 0.08% | 0.98% |
| ICU vital signs | Blood-pressure consistency violations | 0.00% | 0.17% |
| ICU inputs | Medication state change outliers | 1.9% | 1.0% |
| Hospital medications | Medication state change outliers | 0.0% | 0.8% |
| Hospital labs | Range-support violations | 0.23% | 1.55% |
| Hospital labs | One-step jump violations | 2.69% | 4.58% |

## A.3 Comparison to DT-GPT

To broaden baseline coverage for our development tasks, we added a restricted comparison against DT-GPT (Makarov et al., 2025), a recent GPT-style EHR forecasting model. Because DT-GPT focuses on forecasting magnesium, respiratory rate, and oxygen saturation only on MIMIC-IV, we compare on these shared variables. Being restricted to the shared ICU forecasting targets, this comparison is intended as an additional comparison point for clinical variable forecasting. Table 9 reports mean absolute error (MAE) on the three shared variables. EHR2Path performs better on the sparse laboratory target magnesium, while reaching comparable results on the two more frequently measured vital-sign targets, respiratory rate and oxygen saturation, despite being designed for broader heterogeneous trajectory forecasting rather than optimization for only these three variables.

Table 9: Comparison to DT-GPT on the shared MIMIC-IV ICU forecasting targets.

| Variable | DT-GPT MAE | EHR2Path MAE |
|---|---|---|
| Magnesium | 0.12 | **0.09** |
| Respiratory Rate | 0.12 | 0.12 |
| $O_2$ Saturation | 0.14 | 0.14 |

### A.4 Subgroup Heterogeneity Analysis

To examine whether performance changes substantially across patient subgroups, we evaluated `E2P-S+T` on three subgroup axes: sex, age, and stay-level condition rarity. We use the same task definitions, metrics and test sets as for the main simulation results in table 2. Sex and age were derived from MIMIC-IV demographics. Stay-level rarity was defined using the prevalence of a stay's discharge ICD categories in the training set; within each task cohort, samples were then split into rare and common halves using a task-local median split of this global rarity score to ensure balanced support.

Overall, the subgroup analysis shows stable performance across analysed patient subgroups. Across sex and stay-level rarity, performance remains broadly similar for the main trajectory-development tasks and binary outcome predictions, without evidence of systematic degradation in any single subgroup. Age shows somewhat larger heterogeneity, most clearly for ED admission and for value-accuracy metrics, although the latter are partially influenced by low subgroup support in some tasks. The ICU development tasks remain comparatively stable.

Table 10: Subgroup results by sex for `E2P-S+T`.

| Task | Metric | all | male | female |
|---|---|---|---|---|
| Hospital Discharge Diagnosis | Macro F1 | 0.50 [0.48, 0.52] | 0.43 [0.41, 0.45] | 0.51 [0.48, 0.54] |
| ED Admission Prediction | Accuracy | 0.68 [0.68, 0.68] | 0.67 [0.66, 0.68] | 0.69 [0.68, 0.71] |
| ED Admission Prediction | Sensitivity | 0.63 [0.63, 0.63] | 0.63 [0.61, 0.66] | 0.63 [0.63, 0.63] |
| ED Admission Prediction | Specificity | 0.73 [0.73, 0.73] | 0.70 [0.70, 0.70] | 0.76 [0.74, 0.78] |
| ED Discharge Diagnosis | Macro F1 | 0.38 [0.33, 0.41] | 0.34 [0.28, 0.38] | 0.36 [0.30, 0.39] |
| ED Vital Sign Development | Event F1 | 0.57 [0.54, 0.60] | 0.58 [0.54, 0.63] | 0.56 [0.52, 0.60] |
| ED Vital Sign Development | Value Acc. | 0.51 [0.44, 0.57] | 0.50 [0.39, 0.61] | 0.51 [0.42, 0.60] |
| ED Vital Sign Development | Value MAE ↓ | 0.10 [0.10, 0.11] | 0.10 [0.10, 0.11] | 0.10 [0.10, 0.11] |
| Hospital Lab Value Development | Event F1 | 0.23 [0.19, 0.27] | 0.23 [0.17, 0.29] | 0.22 [0.17, 0.28] |
| Hospital Lab Value Development | Value Acc. | 0.44 [0.11, 0.82] | 0.62 [0.00, 1.00] | 0.33 [0.00, 0.83] |
| Hospital Lab Value Development | Value MAE ↓ | 0.08 [0.08, 0.09] | 0.08 [0.07, 0.09] | 0.08 [0.08, 0.09] |
| Hospital Medication Development | Event F1 | 0.83 [0.80, 0.85] | 0.83 [0.79, 0.87] | 0.82 [0.79, 0.86] |
| ICU Imminent Discharge | Accuracy | 0.69 [0.69, 0.69] | 0.68 [0.68, 0.69] | 0.70 [0.69, 0.71] |
| ICU Imminent Discharge | Sensitivity | 0.82 [0.82, 0.82] | 0.82 [0.81, 0.84] | 0.82 [0.82, 0.82] |
| ICU Imminent Discharge | Specificity | 0.56 [0.56, 0.56] | 0.55 [0.55, 0.55] | 0.59 [0.56, 0.61] |
| ICU Input Development | Event F1 | 0.84 [0.82, 0.87] | 0.85 [0.82, 0.88] | 0.83 [0.79, 0.87] |
| ICU Imminent Mortality | Accuracy | 0.57 [0.57, 0.57] | 0.58 [0.58, 0.58] | 0.56 [0.55, 0.57] |
| ICU Imminent Mortality | Sensitivity | 0.16 [0.16, 0.16] | 0.17 [0.17, 0.17] | 0.16 [0.14, 0.18] |
| ICU Imminent Mortality | Specificity | 0.98 [0.98, 0.98] | 1.00 [1.00, 1.00] | 0.96 [0.96, 0.96] |
| ICU Vital Sign Development | Event F1 | 0.70 [0.68, 0.73] | 0.71 [0.68, 0.73] | 0.70 [0.66, 0.73] |
| ICU Vital Sign Development | Value Acc. | 0.79 [0.77, 0.81] | 0.79 [0.76, 0.82] | 0.79 [0.75, 0.83] |
| ICU Vital Sign Development | Value MAE ↓ | 0.08 [0.08, 0.08] | 0.08 [0.07, 0.08] | 0.08 [0.08, 0.09] |

Table 11: Subgroup results by age tertile for `E2P-S+T`.

| Task | Metric | all | young | middle | old |
|---|---|---|---|---|---|
| Hospital Discharge Diagnosis | Macro F1 | 0.50 [0.48, 0.52] | 0.46 [0.42, 0.49] | 0.45 [0.42, 0.48] | 0.44 [0.40, 0.47] |
| ED Admission Prediction | Accuracy | 0.68 [0.68, 0.68] | 0.60 [0.55, 0.64] | 0.70 [0.69, 0.71] | 0.62 [0.57, 0.66] |
| ED Admission Prediction | Sensitivity | 0.63 [0.63, 0.63] | 0.43 [0.43, 0.43] | 0.61 [0.59, 0.63] | 0.73 [0.65, 0.81] |
| ED Admission Prediction | Specificity | 0.73 [0.73, 0.73] | 0.78 [0.67, 0.86] | 0.79 [0.79, 0.79] | 0.50 [0.50, 0.50] |

| Task | Metric | all | young | middle | old |
|------|--------|-----|-------|--------|-----|
| ED Discharge Diagnosis | Macro F1 | 0.38 [0.33, 0.41] | 0.40 [0.28, 0.44] | 0.30 [0.22, 0.35] | 0.35 [0.28, 0.40] |
| ED Vital Sign Development | Event F1 | 0.57 [0.54, 0.60] | 0.56 [0.51, 0.62] | 0.58 [0.53, 0.63] | 0.58 [0.52, 0.62] |
| ED Vital Sign Development | Value Acc. | 0.51 [0.44, 0.57] | 0.44 [0.33, 0.55] | 0.43 [0.33, 0.54] | 0.61 [0.51, 0.73] |
| ED Vital Sign Development | Value MAE ↓ | 0.10 [0.10, 0.11] | 0.10 [0.10, 0.11] | 0.10 [0.09, 0.10] | 0.11 [0.10, 0.12] |
| Hospital Lab Value Development | Event F1 | 0.23 [0.19, 0.27] | 0.19 [0.11, 0.29] | 0.24 [0.18, 0.31] | 0.23 [0.17, 0.29] |
| Hospital Lab Value Development | Value Acc. | 0.44 [0.11, 0.82] | 1.00 [1.00, 1.00] | 0.00 [0.00, 0.00] | 0.42 [0.00, 0.90] |
| Hospital Lab Value Development | Value MAE ↓ | 0.08 [0.08, 0.09] | 0.08 [0.07, 0.09] | 0.08 [0.07, 0.08] | 0.09 [0.08, 0.10] |
| Hospital Medication Development | Event F1 | 0.83 [0.80, 0.85] | 0.78 [0.72, 0.83] | 0.83 [0.78, 0.87] | 0.85 [0.81, 0.89] |
| ICU Imminent Discharge | Accuracy | 0.69 [0.69, 0.69] | 0.70 [0.70, 0.72] | 0.67 [0.66, 0.68] | 0.70 [0.69, 0.70] |
| ICU Imminent Discharge | Sensitivity | 0.82 [0.82, 0.82] | 0.86 [0.85, 0.89] | 0.79 [0.79, 0.79] | 0.79 [0.78, 0.81] |
| ICU Imminent Discharge | Specificity | 0.56 [0.56, 0.56] | 0.54 [0.54, 0.54] | 0.55 [0.52, 0.57] | 0.60 [0.60, 0.60] |
| ICU Input Development | Event F1 | 0.84 [0.82, 0.87] | 0.86 [0.83, 0.90] | 0.80 [0.74, 0.84] | 0.86 [0.83, 0.89] |
| ICU Imminent Mortality | Accuracy | 0.57 [0.57, 0.57] | 0.58 [0.58, 0.59] | 0.55 [0.55, 0.56] | 0.58 [0.55, 0.62] |
| ICU Imminent Mortality | Sensitivity | 0.16 [0.16, 0.16] | 0.17 [0.17, 0.17] | 0.14 [0.14, 0.14] | 0.17 [0.10, 0.25] |
| ICU Imminent Mortality | Specificity | 0.98 [0.98, 0.98] | 0.99 [0.99, 1.00] | 0.97 [0.95, 0.99] | 1.00 [1.00, 1.00] |
| ICU Vital Sign Development | Event F1 | 0.70 [0.68, 0.73] | 0.69 [0.65, 0.73] | 0.68 [0.64, 0.73] | 0.73 [0.69, 0.77] |
| ICU Vital Sign Development | Value Acc. | 0.79 [0.77, 0.81] | 0.79 [0.75, 0.82] | 0.80 [0.75, 0.84] | 0.79 [0.76, 0.83] |
| ICU Vital Sign Development | Value MAE ↓ | 0.08 [0.08, 0.08] | 0.08 [0.08, 0.08] | 0.08 [0.07, 0.08] | 0.08 [0.07, 0.08] |

Table 12: Subgroup results by stay-level rarity for `E2P-S+T`. Rare/common groups are defined by a task-local median split of a global ICD-based rarity score. Diagnosis-task subgroup comparisons are less directly comparable because the rare and common groups differ substantially in active ICD label support.

| Task | Metric | all | rare condition | common condition |
|------|--------|-----|----------------|------------------|
| Hospital Discharge Diagnosis | Macro F1 | 0.50 [0.48, 0.52] | 0.53 [0.50, 0.55] | 0.57 [0.54, 0.61] |
| ED Admission Prediction | Accuracy | 0.68 [0.68, 0.68] | 0.72 [0.72, 0.72] | 0.65 [0.65, 0.65] |
| ED Admission Prediction | Sensitivity | 0.63 [0.63, 0.63] | 0.71 [0.71, 0.71] | 0.55 [0.55, 0.55] |
| ED Admission Prediction | Specificity | 0.73 [0.73, 0.73] | 0.72 [0.72, 0.72] | 0.74 [0.74, 0.74] |
| ED Discharge Diagnosis | Macro F1 | 0.38 [0.33, 0.41] | 0.48 [0.41, 0.53] | 0.64 [0.55, 0.71] |
| ED Vital Sign Development | Event F1 | 0.57 [0.54, 0.60] | 0.57 [0.53, 0.61] | 0.58 [0.53, 0.62] |
| ED Vital Sign Development | Value Acc. | 0.51 [0.44, 0.57] | 0.47 [0.38, 0.56] | 0.54 [0.45, 0.63] |
| ED Vital Sign Development | Value MAE ↓ | 0.10 [0.10, 0.11] | 0.10 [0.10, 0.11] | 0.10 [0.10, 0.11] |
| Hospital Lab Value Development | Event F1 | 0.23 [0.19, 0.27] | 0.19 [0.14, 0.24] | 0.27 [0.21, 0.33] |
| Hospital Lab Value Development | Value Acc. | 0.44 [0.11, 0.82] | 0.70 [0.00, 1.00] | 0.23 [0.00, 0.50] |
| Hospital Lab Value Development | Value MAE ↓ | 0.08 [0.08, 0.09] | 0.08 [0.08, 0.09] | 0.08 [0.07, 0.09] |
| Hospital Medication Development | Event F1 | 0.83 [0.80, 0.85] | 0.84 [0.81, 0.88] | 0.81 [0.77, 0.84] |
| ICU Imminent Discharge | Accuracy | 0.69 [0.69, 0.69] | 0.69 [0.68, 0.70] | 0.69 [0.68, 0.70] |
| ICU Imminent Discharge | Sensitivity | 0.82 [0.82, 0.82] | 0.83 [0.83, 0.83] | 0.81 [0.80, 0.82] |
| ICU Imminent Discharge | Specificity | 0.56 [0.56, 0.56] | 0.55 [0.53, 0.57] | 0.57 [0.57, 0.57] |
| ICU Input Development | Event F1 | 0.84 [0.82, 0.87] | 0.85 [0.81, 0.88] | 0.84 [0.80, 0.87] |
| ICU Imminent Mortality | Accuracy | 0.57 [0.57, 0.57] | 0.60 [0.58, 0.61] | 0.55 [0.55, 0.56] |
| ICU Imminent Mortality | Sensitivity | 0.16 [0.16, 0.16] | 0.19 [0.17, 0.21] | 0.13 [0.13, 0.13] |
| ICU Imminent Mortality | Specificity | 0.98 [0.98, 0.98] | 1.00 [1.00, 1.00] | 0.97 [0.96, 0.98] |
| ICU Vital Sign Development | Event F1 | 0.70 [0.68, 0.73] | 0.70 [0.67, 0.74] | 0.70 [0.66, 0.73] |
| ICU Vital Sign Development | Value Acc. | 0.79 [0.77, 0.81] | 0.78 [0.75, 0.81] | 0.80 [0.77, 0.84] |
| ICU Vital Sign Development | Value MAE ↓ | 0.08 [0.08, 0.08] | 0.08 [0.08, 0.08] | 0.08 [0.08, 0.08] |

## A.5 Qualitative Examples of Multi-Step Simulation

To qualitatively illustrate the behavior of EHR2Path-S+T under iterative rollout, we show qualitative examples from different task settings. Figure 7 shows an example of the predicted and real development of different vital signs of a patient, predicted by the combined model EHR2Path-S+T. Figure 8 complements this with four additional multi-step examples from two clinically interpretable feature sets: ICU vasopressor treatment together with blood-pressure trajectories, and a partial hospital renal panel consisting of creatinine and urea nitrogen.

Across these examples, the generated rollouts remain temporally coherent and clinically plausible over multiple forecast steps, without drifting into physiologically nonsensical states. The vasopressor examples illustrate

that treatment changes can occasionally be predicted with a small temporal delay, while still capturing the correct overall intervention pattern and its relationship to blood-pressure dynamics. The renal-panel examples show that the model often captures the overall direction of change, but may produce smoother trajectories that underestimate sharper deviations.

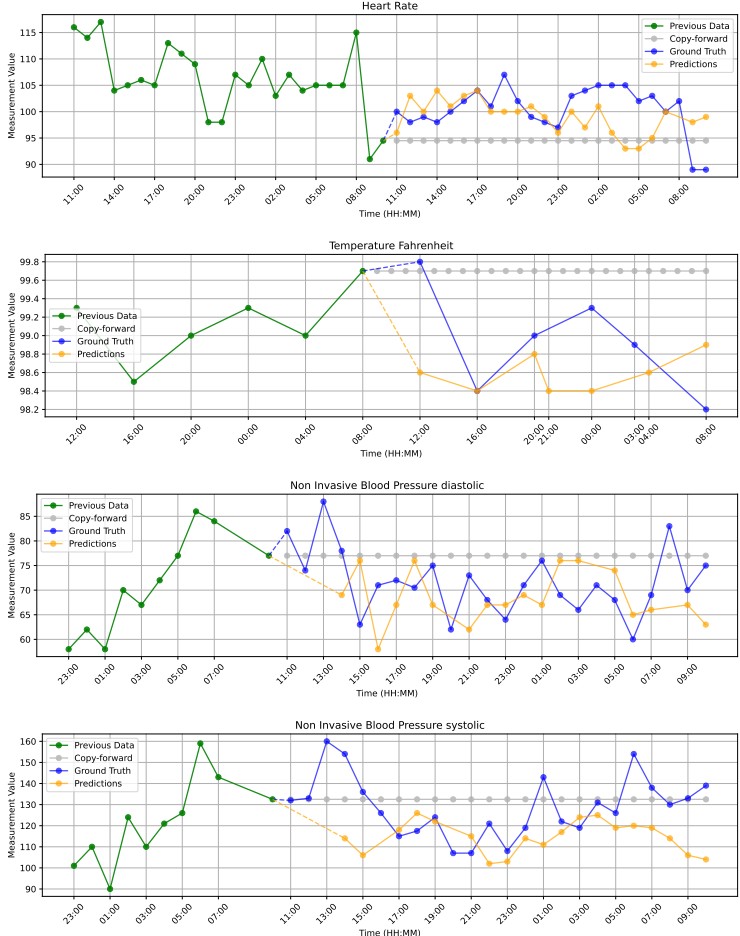

Figure 7: Qualitative example of the ICU Vital Sign Simulation over 24 hours, showing ground-truth (blue), prediction (yellow), 24h prior development (green) and copy-forward reference (grey) of Temperature, Heart Rate and diastolic and systolic blood pressure of the same patient. The rollout remains temporally coherent across multiple vital signs, while correctly estimating their developments.

## A.6  Qualitative analysis of summary-token attention

Figure 9 shows qualitative attention visualizations for four representative section types: procedures, medications, vital signs, and lab results. For each case, we compute attention from each of the eight summary tokens to structured clinical input spans, where each span corresponds to a field together with its associated values. For readability, each heatmap shows the union of the top-5 attended spans across the summary tokens. Green starred labels indicate spans that are also reflected in the expected next-hour output.

The visualizations show two key patterns. First, different summary tokens focus on different parts of the input, suggesting that the compressed representation is distributed across tokens rather than collapsing onto the same spans. Second, the dominant attended spans frequently overlap with information that appears in the next-hour output, indicating that the bottleneck emphasizes forecast-relevant content, which is exactly what the summary-model training objective is designed to encourage. This is especially clear in the procedure and medication cases, where the most strongly attended spans correspond closely to procedures or medications

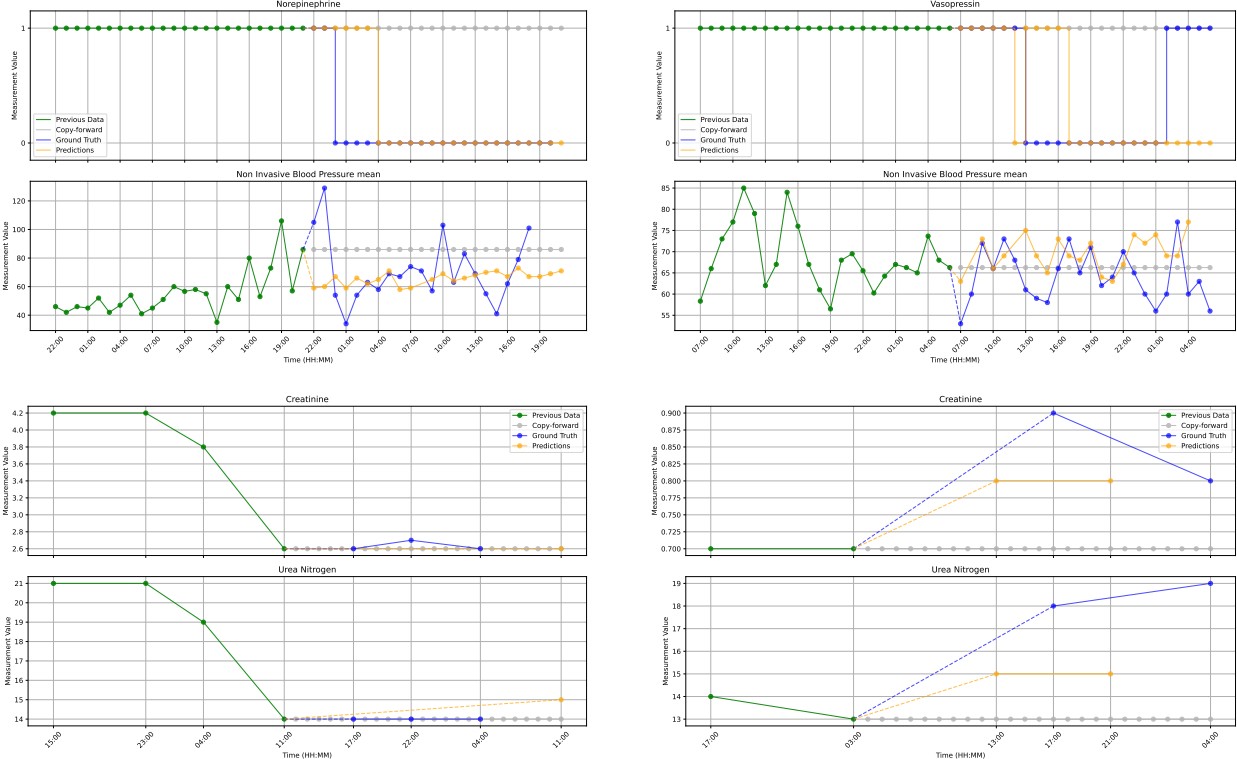

Figure 8: Qualitative example of multi-step simulation. Top row: two ICU vasopressor - blood-pressure examples, where the model captures the overall intervention pattern and associated blood-pressure development, with occasional small delays in treatment timing. Bottom row: two partial hospital renal-panel examples (creatinine and urea nitrogen), showing the model follows the overall lab trajectory while in one example tending to produce smoother forecasts than the ground truth.

that recur in the next-hour output. In the more complex vital-sign and lab-result cases, many of the top-attended fields also appear in the next-hour output, while additional attention is allocated to other common measurements present in the input. Taken together, these examples support that the bottleneck learns a non-trivial, task-aligned compression.

### A.7 Example of our structured text representation

In fig. 10 and fig. 11, we show examples for the textual input and output representation for one time-point of a single patient.

### A.8 Implementation Details

Our model is implemented in PyTorch and based on the `Qwen-0.5b` model Yang et al. (2024), a compact LLM balancing capacity and efficiency, provided by unsloth Daniel Han & team (2023) under the apache-2.0 license (`https://huggingface.co/unsloth/Qwen2-0.5B-Instruct-bnb-4bit`). We chose this compact backbone to support the resource-intensive setting combining long-context inputs and iterative rollout evaluation, while also allowing us to assess whether the proposed representation and summarization approach is effective at a modest model scale. We train using a next-token objective with LoRA Hu et al. (2022) on structured text outputs (section 3.1), optimized with AdamW ($1 \times 10^{-4}$ learning rate). At inference, we use the default Qwen decoding parameters ($temperature = 0.7, top_k = 20, top_p = 0.8$). Training runs on a single NVIDIA A40 GPU (48 GB) with a batch size of 8 for one epoch, covering one million time points. The summary-based model trains in 1.5 days, while the 24h text-based model and the combined model require approximately

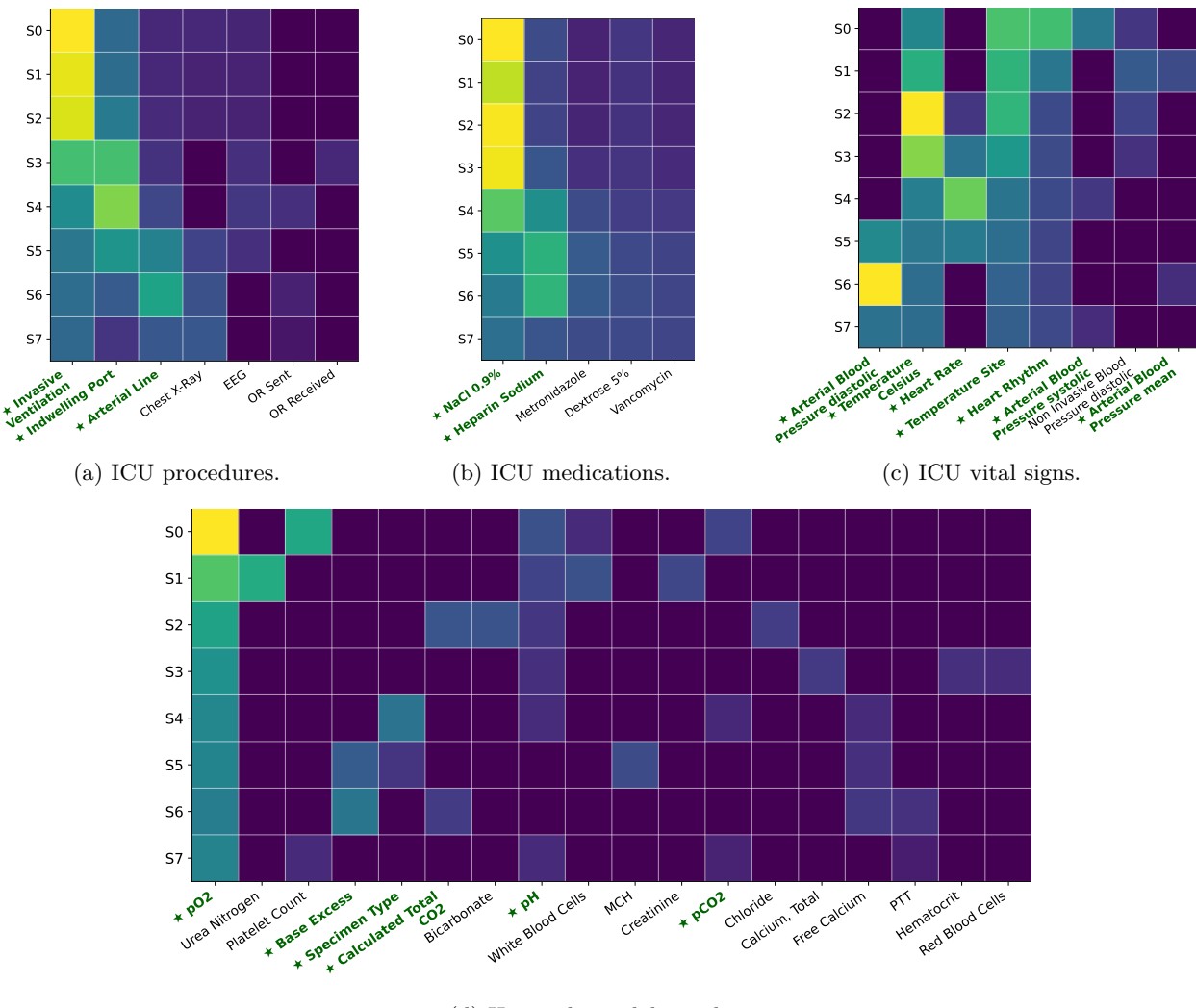

(a) ICU procedures.  (b) ICU medications.  (c) ICU vital signs.

(d) Hospital-stay lab results.

Figure 9: Span-level attention from the eight summary tokens for four summary sections. Each cell reports the fraction of a summary token's normalized input-directed attention mass that falls within the corresponding field span. Green-starred labels indicate spans that are also present in the expected next-hour output.

4.0 days. The summarization model, trained separately on one million section samples, also completes in 4.0 days. All pathway models have soft 4000-token limit, realized by truncating text based on approximate token counts estimated from character counts. In the summary-based model, each section of max. 5000 tokens is compressed into 8 summary embeddings, achieving up to 625x compression. For the summary+text model, we leverage the summary-based model as starting point and fine-tune it using curriculum learning to leverage additional text inputs. During fine-tuning we randomly drop the text or summary inputs in 1/3rd of the training samples each. This incentivizes the model to leverage both input types. At inference time our simulation takes on average 3.5 seconds per simulated hour of data on a single GPU. For all baselines, we use the original training and inference parameters. For all models, results are reported from a single run with bootstrapped 95% confidence intervals.

## A.9 Detailed Task Descriptions

**ED Tasks:**
*Vital Sign Development*: Predicts the progression of vital signs over 24 hours or until ED discharge, using

data up to a random time point in the ED stay.

*Admission Prediction*: Determines whether an ED patient will be hospitalized or discharged home, using all static and dynamic data available at ED admission, simulating until the ED stay ends. The ground-truth label is positive if the final ED disposition is hospital admission. The rollout-derived prediction is positive if the generated ED disposition is `ADMITTED`, and negative if the generated terminal ED disposition is `DISCHARGED` or `DIED`. A fixed maximum-step guard is additionally used as fallback; if reached, the simulation is stopped and the prediction is assigned a negative label, but this fallback was not triggered in our experiments.

*Discharge Diagnosis*: Forecasts ICD categories at ED discharge (multi-label prediction over 18 categories as defined by Slee (1978)), using all ED data up to that point. The LOS indicator is set to zero for direct prediction of disposition and ICD categories.

**Hospital Tasks:**

*Medication Development*: Predicts administered medications over 24 hours or until hospital discharge, using data up to a random time point in the hospital ward.

*Lab Value Development*: Predicts performed lab tests and their results over 24 hours or until discharge, using data up to a random time point in the hospital ward.

*Discharge Diagnosis*: Forecasts ICD categories at hospital discharge (multi-label prediction over 18 categories as defined by Slee (1978)), using all patient data up to that point. The LOS indicator is set to zero for direct prediction.

**ICU Tasks:**

*Vital Sign Development*: Models ICU vital sign trajectories over 24 hours or until ICU discharge, using data up to a random time point in the ICU.

*Input Development*: Predicts administration of inputs (e.g., transfusions, medications) over 24 hours or until ICU discharge, using data up to a random time point in the ICU.

*Imminent Mortality*: Assesses whether a patient will die within 24 hours, using data up to a random time point in the ICU, with simulation continuing until discharge or death. The ground-truth label is positive if death occurs within the 24-hour prediction window. The rollout-derived prediction is positive if the generated trajectory contains a death disposition within this horizon, and negative otherwise.

*Imminent Discharge*: Predicts whether a patient will be discharged from the ICU within 3 days. Inputs include data up to the end of the first day in the stay, with simulation extending 72 hours or until discharge. The ground-truth label is positive if ICU release occurs within the 72-hour prediction window. The rollout-derived prediction is positive if an ICU release event is generated within 72 hours, and negative otherwise.

For all tasks, rollouts terminate when a relevant terminal state is generated or when a fixed task-specific maximum horizon is reached. Prediction labels are obtained by rule-based parsing of generated structured outputs.

| Task | Task Type | Input Window | Gap Window | Output Window | Rolling/Static |
|------|-----------|--------------|------------|---------------|----------------|
| ED Vital Signs | Development | up to random ED timepoint | 1h | 24h or until ED stay end | Rolling |
| ED Admission | Outcome | up to ED admission | – | ED stay end | Static |
| ED Discharge Diagnosis | Outcome | up to ED discharge | – | Direct prediction | Static |
| Hospital Medications | Development | up to random hospital timepoint | 1h | 24h or until hospital discharge | Rolling |
| Hospital Lab Values | Development | up to random hospital timepoint | 1h | 24h or until hospital discharge | Rolling |
| Hospital Discharge Diagnosis | Outcome | up to discharge | – | Direct prediction | Static |
| ICU Vital Signs | Development | up to random ICU timepoint | 1h | 24h or until ICU discharge | Rolling |
| ICU Inputs | Development | up to random ICU timepoint | 1h | 24h or until ICU discharge | Rolling |
| ICU Imminent Mortality | Outcome | up to random ICU timepoint | 1h | within 24h | Rolling |
| ICU Imminent Discharge | Outcome | first 24h of ICU stay | 1h | within 3 days | Rolling |

Table 13: Details of task definitions, including task type, input/ gap / output window and if the task is rolling or static.

Figure 10: Example of structured input data from a patient record.

```
'Hospital Stay':
  'General': 'ew emer. patient, 57-year old male, insurance: Medicare, white, language:
    english'
  'Patient Location': '175-167: Emergency Department,167-135: Medical Intensive Care
    Unit (MICU),135-0: Coronary Care Unit (CCU)'
  'Care Taker': '169-3: Medical,3-0: Cardiac Medical'
  'Outpatient Measurements':
    'Height (Inches)': '174 days: 71'
  'Lab Results':
    'Lactate (mmol/L, normal range: 0.5-2.0)': '165: 7.8, 164: 7.6, [...] 143/136: 2.2,
      133: 2.1, 122: 2.4, 119: 1.8'
  'Microbiology Growth Results':
    'GRAM STAIN - sputum': '145: no growth, 135: no growth'
  'Prescriptions':
    'sodium polystyrene sulfonate': '141-135'
    'dextrose': '97-79'
    'amiodarone hydrochloride': '97-74'
    [...]
  'Procedures':
    '7 days': 'Insertion of endotracheal tube; Continuous invasive mechanical ventilation
      for 96 consecutive hours or more; [...]'
  'Radiology Notes':
    '175': 'CHEST (PORTABLE AP): Limited study with new increased opacification of
      the left mediastinal contour and left heart border. Left-sided pleural effusion
      or focal consolidation cannot be excluded on this study. There is mild interstitial
      edema.'
'ICU Stay':
  'Stay 0':
    'Medication':
      'Heparin Sodium': '103-0'
    'Output':
      'Foley(ml)': '167-166: 0, 164: 10, 163-160: 0, 159: 40, 158: 15, 156: 24, 153:
        80, 152: 20, 151: 30, [...], 5: 90, 3: 180, 1: 220'
    'Chart Events':
      'Cardiovascular':
        'RLE Color': '165-161: Mottled, 159-41: Normal, 36/33/29/21/17: Normal'
      'RoutineVitalSigns':
        'Arterial Blood Pressure systolic(mmHg)': '165: 71.5, 164: 118, 163: 94, 162:
          89, 161: 92, [...], 11: 113, 10: 99, 9: 110, 7: 114.5,
          6: 106, 5: 104'
      'AdmHistory_FHPA':
        'Unable to assess teaching / learning needs': '130: 1'
      'Respiratory':
        'ETT Location': '166-108: Oral-R, 105-93: Oral-L, 89-77: Oral Center, 73/69/64/60/57:
          Oral Center'
      'Pulmonary':
        'Cough Effort': '167/117/113/109/41: Weak'
      'Skin-Assessment':
        'Braden Nutrition': '166: Probably Inadequate, 165-157: Adequate, 153-145:
          Probably Inadequate, 141-137: Adequate, 134-80: Probably Inadequate, 72:
          Very Poor, 61-50: Probably Inadequate, 45/36/21/8/1: Probably Inadequate'
          [...]
```

(a) Example of an expected output for the next hour during the stay.

```
'Hospital Stay':
  'LOS': '116 hours'
  'Patient Location': 'Coronary Care Unit (CCU)'
  'Care Taker': 'CMED'
  'Prescriptions':
  - 'oxycodone hydrochloride and acetaminophen'
  - 'aspirin'
  - 'clotrimazole'
  [...]
'ICU Stay':
  'Stay 0':
    'LOS': '23 hours'
    'Medication':
    - 'Dextrose 5%'
    - 'Heparin Sodium'
    - 'Insulin - Regular'
    [...]
    'Procedures':
    - 'Multi Lumen'
    'Chart Events':
      'RoutineVitalSigns':
        'Heart Rate': '82.0'
        'Heart Rhythm': '1st AV (First degree AV Block) '
        'Non Invasive Blood Pressure diastolic': '78.0'
        [...]
      'Respiratory':
        'O2 saturation pulseoxymetry': '94.0'
        'Respiratory Rate': '17.0'
```

(b) Example of an expected output for the next hour at the end of the stay.

```
'Hospital Stay':
  'Disposition': 'DISCHARGED'
  'ICD categories': 'respiratory;pregnancy;congenital'
```

Figure 11: Examples of an expected output for the next hour during the stay (top) and at the end of the stay (bottom).

