# OpenReview forum: "EHR2Path: Comprehensive Pathway-Level Modeling of Longitudinal Patient Trajectories from Multimodal Electronic Health Records"
_TMLR — Decision pending for TMLR_

### Review · Reviewer_8zsN · 2026-04-10

**Summary Of Contributions:**

This paper introduces EHR2Path, a framework for forecasting and simulating full in-hospital patient pathways from EHRs. It works by serialising heterogeneous EHR data  (structured codes, labs, vitals, medications, free-text notes, and dense ICU chart events) into a unified text representation, then trains a LLM  to predict the next hourly time-step. A Masked Summarization Bottleneck compresses long patient history into a small set of learned summary tokens via a custom attention mask, achieving huge compression while preserving predictive information. A Length-of-Stay (LOS) indicator is used to help the model converge to terminal states. Evaluation spans 9 tasks on MIMIC-IV (ED, ward, ICU) covering both trajectory simulation and outcome prediction, compared against ETHOS, MEME, and REMed.

Key strengths:
- Broad task coverage across clinical stages;
- The summarization bottleneck is a genuine and potentially generalizable architectural idea;
- Honest evaluation with confidence intervals, a statistical baseline, and a restricted-feature comparison for ETHOS.

Key weaknesses:
- The baseline set is narrow: ETHOS struggles on the full feature space (though it is competitive on its supported subset and on numerical prediction), and the other baselines (MEME, REMed) each cover only a subset of tasks.
- The evaluation is single-dataset (MIMIC-IV only); and
- Clinical validity of simulated trajectories is assessed only via statistical fidelity metrics.

**Audience:**

Yes

**Audience Explanation:**

The idea is not new, but still useful as health trajectory modelling is a very important problem.

**Broader Impact Concerns:**

Not at this stage, but claims on health have major impact, so be conservative.

**Claims And Evidence:**

No

**Claims Explanation:**

The evidence is partially convincing but has several gaps that prevent full confidence in the claims:

- Baseline coverage is too narrow. The trajectory modelling using LSTMs was introduced 10 years ago with DeepCare. Serialisation as introduced 9 years ago with Deepr. Since then, there have been hundreds of papers along the line, especially using autoregressive GPTs instead of RNNs.

- SThe paper evaluates simulated trajectories using event F1 and value MAE, but these do not assess whether generated sequences are clinically plausible. A trajectory could achieve reasonable F1 while producing physiologically impossible vital sign combinations or contradictory medication sequences.

- For a journal paper claiming to establish a foundation for pathway modeling, validation on MIMIC-IV alone  is insufficient. The text serialisation is tightly coupled to MIMIC-IV's 22 specific tables (Table 6), and transferability to other EHR systems, coding standards, or hospital types is untested.

- The existing ablations provide useful evidence that the bottleneck contributes, but do not isolate it from the base model's context capacity. Comparing against Qwen2-0.5B with a longer native context window (4K or 8K tokens) but no bottleneck would clarify whether the bottleneck adds genuine value beyond what simply extending context achieves.

**Requested Changes:**

Critical
=====

- Strengthen baselines.
- Better abliation studies.
- Add clinical plausibility evaluation. Expert clinician review of a sample of simulated trajectories would further strengthen the work.


Recommended
============
- Qwen2-0.5B is very small. Any reason why?
- Rare/severse critical events are important in health. Averaging accuracy on everything isn't.

---

> ### Author Response · Authors · 2026-04-21
> **Official Comment by Authors (Part 1)**
>
> We thank the reviewer for their constructive feedback and for recognizing the relevance of the task, the “Broad task coverage across clinical stages”, the Masked Summarization Bottleneck as “a genuine and potentially generalizable architectural idea”, and our “Honest evaluation with confidence intervals, a statistical baseline, and a restricted-feature comparison”. We have added additional clarifications and experiments below and revised the manuscript accordingly. All changes are highlighted in blue in the revised manuscript.
>
> **1\) “The paper evaluates simulated trajectories using event F1 and value MAE, but these do not assess whether generated sequences are clinically plausible. A trajectory could achieve reasonable F1 while producing physiologically impossible vital sign combinations or contradictory medication sequences. Add clinical plausibility evaluation. Expert clinician review of a sample of simulated trajectories would further strengthen the work.”**
>
> **Response:**
>
> We appreciate this suggestion. To improve the evaluation of clinical plausibility in generated trajectories, we **added an automated clinical plausibility analysis** for generated E2P-T+S rollout windows and compared them to matched real data windows from the test set. The analysis checks four complementary failure modes: values outside the empirical support of the data, unlikely one-step jumps, physiologically inconsistent blood-pressure combinations, and unusually frequent medication state changes.
>
> Concretely, range violations are defined using empirical 1st–99th percentile intervals, jump violations using empirical 99th percentile thresholds of one-step changes, blood-pressure consistency by requiring systolic ≥ diastolic and mean arterial pressure to lie between them when present, and medication instability by flagging rollout windows with unusually many on/off changes.
>
> Across 24 hour simulations of ED vital signs, ICU vital signs, ICU inputs, hospital medications, and hospital labs, the observed violation rates are low and broadly comparable to the matched real windows. **This new analysis supports that the generated trajectories are not only accurate under task metrics, but also clinically and temporally consistent under these empirical checks.**
>
> **Changes in the paper:**
>
> * Revised **Section 5.2 (Patient Trajectory Simulation Results)**: added short description of the main take-aways from the new plausibility analysis
> * Added **Appendix A.2 \+ Table 7:** full Clinical Plausibility Analysis.
> * Revised **Section 5.5 (Limitations and Future Work)**: Discuss clinical validation as important future work.
>
> New Table 7:
>
> Task | Check | Generated | Real
> -|-|-|-
> ED vital signs | Range-support violations | 0.47% | 1.07%
> ED vital signs | One-step jump violations | 0.39% | 1.08%
> ED vital signs | Blood-pressure consistency violations | 0.00% | 0.00%
> ICU vital signs | Range-support violations | 0.28% | 1.69%
> ICU vital signs | One-step jump violations | 0.08% | 0.98%
> ICU vital signs | Blood-pressure consistency violations | 0.00% | 0.17%
> ICU inputs | Medication state change outliers | 1.9% | 1.0%
> Hospital medications | Medication state change outliers | 0.0% | 0.8%
> Hospital labs | Range-support violations | 0.23% | 1.55%
> Hospital labs | One-step jump violations | 2.69% | 4.58%
>
> **2\) “Baseline coverage is too narrow. \[...\] there have been hundreds of papers along the line, especially using autoregressive GPTs instead of RNNs.”**
>
> **Response:**
>
> **We selected our original baselines to match the task setting**: many prior EHR models are outcome-focused or limited to forecasting a small set of continuous variables, and are therefore not directly comparable to our broader heterogeneous trajectory-generation setup. We therefore use ETHOS as the primary baseline for generative trajectory modeling, while Table 2 additionally compares against MEME and ReMed on the more specialized tasks they address.
>
> To further broaden baseline coverage, **we added a restricted comparison against DT-GPT (Makarov et al., 2025\)**, a recent GPT-style EHR time-series forecasting model. Because DT-GPT’s MIMIC-IV ICU evaluation is limited to forecasting magnesium, respiratory rate, and oxygen saturation, a fair comparison is only possible on these shared targets. On this restricted set, EHR2Path performs better on magnesium (MAE 0.09 vs. 0.12) and matches DT-GPT on respiratory rate and oxygen saturation (both 0.12 and 0.14, respectively). This is encouraging, since EHR2Path is designed for substantially broader heterogeneous trajectory forecasting rather than optimization for only these few variables, yet remains competitive with or better than a recent GPT-style forecasting baseline on the directly shared targets.
>
> **Changes in the paper:**
>
> * Added **Appendix A.3 \+ Table 8**: Comparison to DT-GPT.
>
> New Table 8:
>
> Variable|DT-GPT|EHR2Path
> -|-|-
> Magnesium|0.12|**0.09**
> Respiratory Rate|0.12|0.12
> O$\_2$ Saturation|0.14|0.14

---

> > ### Author Response · Authors · 2026-04-21
> > **Official Comment by Authors (Part 2)**
> >
> > **3\) “The existing ablations provide useful evidence that the bottleneck contributes, but do not isolate it from the base model's context capacity. Comparing against Qwen2-0.5B with a longer native context window (4K or 8K tokens) but no bottleneck would clarify whether the bottleneck adds genuine value beyond what simply extending context achieves.”**
> >
> > **Response:**
> >
> > We thank the reviewer for raising this important point. In revisiting this comparison, we realized that Table 1 in the original manuscript incorrectly suggested that E2P-T+S used more total input tokens than the text-only baseline. In the actual implementation, this is not the case: **all pathway variants use the same soft 4000-token input limit**, and in E2P-T+S the summary tokens replace part of the raw-text budget rather than increasing total prompt length. We have corrected Table 1 and adapted the results section to more precisely discuss our fixed-budget setting: under the same 4000-token budget, the summary-based variant represents substantially more longitudinal history by trading some recent raw text for compressed long-history context. **These results support that, in our fixed-budget setting, using part of the budget for learned summary tokens is more effective than allocating the full budget only to recent raw text.** We have corrected Table 1 and made the setting more clear in the discussion.
> >
> > **To provide additional analysis of the role of the bottleneck, we added a qualitative analysis of the learned summary-token attention.** For representative procedure, medication, vital-sign, and lab-result sections, we visualize which structured input spans receive the strongest attention from each of the eight summary tokens. Two key patterns are visible. First, different summary tokens focus on different parts of the input, suggesting that the compressed representation is distributed across tokens rather than collapsing onto the same spans. Second, the dominant attended spans frequently overlap with information that appears in the next-hour output, indicating that the bottleneck emphasizes forecast-relevant content, which is exactly what the summary-model training objective is designed to encourage. Taken together, **these results support that the bottleneck learns a non-trivial, task-aligned compression under the fixed input budget**, rather than acting as a pure context extension.
> >
> > **Changes in the paper:**
> >
> > * Corrected **Table 1**: E2P-T+S uses the same intended input budget as the text-only variant. Small differences in exact lengths arise from approximate token-count-based truncation.
> > * Revised **Section 5.1 (Next Timestep Prediction Results)**: more explicitly describe and interpret the fixed-budget comparison.
> > * Added clarification in **Appendix A.9 (Implementation Details)**: all pathway models use a soft 4000-token limit realized via approximate token-count-based truncation.
> > * Added **Appendix A.7**: Attention Visualizations of summary token creation
> >
> > **4\) “For a journal paper claiming to establish a foundation for pathway modeling, validation on MIMIC-IV alone is insufficient. The text serialisation is tightly coupled to MIMIC-IV's 22 specific tables (Table 6), and transferability to other EHR systems, coding standards, or hospital types is untested.”**
> >
> > **Response:**
> >
> > We agree that validation on MIMIC-IV alone does not establish transferability across healthcare systems, which is why **we acknowledged this in the limitation section of the original manuscript**. We also agree the framing can be refined in some places and have **revised the manuscript to directly address the mentioned claims**, clarifying that the paper demonstrates retrospective pathway modeling on long, multimodal MIMIC-IV trajectories. At the same time, we believe the current evaluation remains valuable because MIMIC-IV is a large, heterogeneous, and clinically realistic benchmark, allowing us to test the feasibility of pathway-level modeling under broad multimodal and long-context conditions.
> >
> > **Changes in the paper:**
> >
> > - We adapted the **title, abstract and conclusion** to better reflect our intended scope
> > - We **changed the word “scalable”** throughout the paper to “pathway-level” in order to clearly show that we refer to scalability with respect to the amount, heterogeneity, and temporal extent of the considered EHR data, rather than demonstrated scalability across healthcare systems.
> > - Revised **Section 5.5 (Limitations and Future Work):** Discuss external and prospective validation as important future work.

---

> ### Author Response · Authors · 2026-04-21
> **Official Comment by Authors (Part 3)**
>
> **5\) “Rare/severse critical events are important in health. Averaging accuracy on everything isn't.”**
>
> **Response:**
>
> We agree that rare and clinically severe events are particularly important to evaluate. In the original submission, we already accounted for rarity during training through importance-weighted sampling and oversampling of critical transitions, and for event prediction we reported macro F1 to reduce dominance by common events. **For the binary outcome prediction tasks**, where the main paper previously reported balanced accuracy for sample efficiency, we **now additionally report sensitivity and specificity**. These metrics make the false-negative / false-positive trade-off explicit and provide a more informative view of performance on rare critical outcomes such as ICU mortality and imminent discharge. The updated results show that the different model variants often trade off sensitivity and specificity differently, while the strongest overall models tend to achieve the most balanced operating points.
>
> **Changes in the paper:**
>
> * **Table 2 and Table 3**: added sensitivity and specificity to binary tasks.
> * Revised **Section 5.2 (Patient Trajectory Simulation Results)**: discuss sensitivity/specificity trade-offs for ED admission, ICU imminent mortality, and ICU imminent discharge.
> * Revised **Section 5.3 (EHR2Path as a Foundation Model for Outcome Prediction)**: discuss sensitivity/specificity trade-offs for the fine-tuned outcome models.
>
> Adapted tasks in Table 2:
> Task | Metric | ETHOS | E2P-T | E2P-S | E2P-S+T
> -|-|-|-|-|-
> ED Admission Prediction | Acc. | n/a | 0.67 (0.63,0.71) | 0.63 (0.59,0.67) | **0.68** (0.64,0.72)
> ED Admission Prediction | Sens. | n/a | 0.58 (0.52,0.64) | 0.48 (0.42,0.54) | **0.63** (0.57,0.69)
> ED Admission Prediction | Spec. | n/a | **0.80** (0.76,0.85) | 0.78 (0.73,0.83) | 0.73 (0.68,0.78)
> ICU Imminent Mortality | Acc. | **0.61** (0.57,0.65) | 0.53 (0.49,0.58) | 0.50 (0.46,0.54) | 0.57 (0.53,0.62)
> ICU Imminent Mortality | Sens. | **0.23** (0.18,0.29) | 0.08 (0.05,0.12) | 0.00 (0.00,0.00) | 0.16 (0.12,0.21)
> ICU Imminent Mortality | Spec. | 0.99 (0.97,1.00) | **1.00** (1.00,1.00) | **1.00** (1.00,1.00) | 0.98 (0.97,1.00)
> ICU Imminent Discharge | Acc. | 0.63 (0.51,0.74) | 0.63 (0.59,0.67) | 0.57 (0.53,0.61) | **0.69** (0.65,0.73)
> ICU Imminent Discharge | Sens. | 0.71 (0.54,0.87) | 0.46 (0.40,0.52) | 0.27 (0.21,0.33) | **0.82** (0.77,0.86)
> ICU Imminent Discharge | Spec. | 0.56 (0.38,0.72) | 0.70 (0.64,0.76) | **0.88** (0.83,0.92) | 0.56 (0.50,0.62)
>
> Adapted tasks in Table 3:
>
> Task | Metric | MEME | REMed | E2P-FT-P | E2P-FT-O
> -|-|-|-|-|-
> ED Admission Prediction | Acc. | 0.67 (0.63,0.71) | n/a | 0.73 (0.69,0.77) | **0.74** (0.70,0.77)
> ED Admission Prediction | Sens. | 0.59 (0.53,0.65) | n/a | 0.72 (0.67,0.78) | **0.74** (0.68,0.79)
> ED Admission Prediction | Spec. | **0.76** (0.70,0.81) | n/a | 0.73 (0.68,0.78) | 0.74 (0.68,0.79)
> ICU Imminent Mortality | Acc. | n/a | 0.71 (0.60,0.80) | 0.77 (0.74,0.81) | **0.83** (0.80,0.86)
> ICU Imminent Mortality | Sens. | n/a | 0.41 (0.25,0.57) | 0.64 (0.59,0.70) | **0.72** (0.67,0.78)
> ICU Imminent Mortality | Spec. | n/a | **1.00** (1.00,1.00) | 0.90 (0.87,0.94) | 0.94 (0.91,0.97)
> ICU Imminent Discharge | Acc. | n/a | **0.82** (0.78,0.85) | 0.69 (0.65,0.73) | 0.76 (0.72,0.79)
> ICU Imminent Discharge | Sens. | n/a | **0.85** (0.80,0.90) | 0.80 (0.75,0.85) | 0.78 (0.72,0.83)
> ICU Imminent Discharge | Spec. | n/a | **0.81** (0.77,0.85) | 0.58 (0.52,0.64) | 0.74 (0.69,0.80)
>
> **6\) “Qwen2-0.5B is very small. Any reason why?”**
>
> **Response:**
>
> Thank you for this question. We used Qwen2-0.5B because our experiments involve long contexts, iterative simulation, and several model variants, making larger backbones difficult to support within our compute budget. We also believe it is useful to show that the proposed framework already provides value at a modest model scale. We have expanded the implementation details to make this rationale explicit. Given more compute budget, bigger models could lead to improved results.
>
> **Changes in the paper:**
>
> * Revised **A.9 Implementation Details:** added rationale for choosing Qwen2-0.5B

---

### Review · Reviewer_jUEC · 2026-04-13

**Summary Of Contributions:**

The paper proposes EHR2Path, a system for predicting and simulating full patient trajectories during hospital stays using transformer LLMs applied to diverse EHR data. The key contributions are: a unified text representation of multimodal EHR data (vitals, labs, notes, medications, procedures); a Masked Summarization Bottleneck that compresses history into summary tokens while keeping recent context as text; and evaluation on nine tasks across ED, ward, and ICU using MIMIC-IV.

Strengths: The scope of data integration is comprehensive. Most prior work picks a subset of structured codes or focuses on narrow outcomes. Here the authors are trying to handle the messy reality of what's actually in EHRs. The iterative simulation evaluation is also more interesting than the typical single-step outcome prediction task. The text representation represents clinical semantics reasonably well.

Weaknesses: The technical novelty is limited. The summarization bottleneck is basically just a custom attention mask that forces outputs to attend only to summary tokens. This is intuitive but not particularly novel. It's unclear why this is better than just using established efficient attention mechanisms. The information bottleneck framing feels like post hoc theoretical justification rather than something that guided the design. The experimental validation is narrow, restricted entirely to MIMIC-IV. For a paper claiming "scalable" modeling this is a significant limitation. The comparison to ETHOS is also problematic. When they restrict to ETHOS-compatible features, the gap shrinks substantially, which suggests they're partly just winning by incorporating more data rather than better methods.

**Additional Comments:**

The structured text representation makes sense but it's verbose. An average hour is 380 tokens which adds up. Did you consider more compact representations? Is this formatting actually efficient for the LLM or are you wasting tokens?

Figure 4 summarizing task coverage is nice but actually hard to interpret because you're mixing task types. ED diagnosis prediction is a different kind of task than vital sign forecasting. Mixing them in one visualization obscures more than it reveals.

**Audience:**

Yes

**Audience Explanation:**

Yes, this is in an active research area and the healthcare ML community would pay attention. But I'd caveat that the interest is probably limited to researchers already working on EHR modeling specifically. This isn't a general ML contribution. In my opinion, the technical innovations aren't strong enough to interest the broader TMLR audience.

**Broader Impact Concerns:**

It's critical to make claims about clinical decision support with clinical validation. The system makes predictions but author(s) haven't shown they're useful or safe to use. Iterative simulation compounds errors, and author(s) haven't analyzed what happens when the model generates inaccurate trajectories. How would a clinician know to not trust bad predictions?

Also, talking about sensitive patient data: Even though the dataset public MIMIC, I encourage the author(s) to discuss privacy considerations for anyone who wants to deploy or adapt this to real hospitals. What validation do they need to do? What are they responsible for?

**Claims And Evidence:**

Yes

**Claims Explanation:**

The scalability claim doesn't hold up. One dataset is not scalable validation. The authors never test on any other healthcare system, patient population, or EHR structure. All their results are MIMIC-IV only. Yes, they integrated a lot of features, but that's not the same as demonstrating the approach would work elsewhere.

The comparison against baselines is also problematic in a few ways. ETHOS wasn't designed for the full feature set EHR2Path uses. When they do the fair comparison in Table 1 by restricting to ETHOS-compatible features, ETHOS gets 0.003 to 0.04 event F1 while E2P-S+T gets 0.12 to 0.24. That's still a gap but much smaller than the headline numbers. They should be clearer that they're comparing different things in the main table.

The Masked Summarization Bottleneck is presented as an important contribution but there's minimal evidence it's actually necessary or optimal. Table 5 shows bottleneck size effects but only using validation loss, not downstream task performance. We don't know if the summary tokens are capturing meaningful clinical patterns or just fitting whatever random initialization works. There's no visualization, no analysis of what gets compressed away. They don't compare to simpler alternatives like just truncating history or using sparse attention.

Last but not least, the paper needs clinical validation. They predict "next hour vital signs" and "next hour medications" as if these are meaningful clinical objectives, but the paper also needs to answer whether clinicians care about these predictions or would use them. The abstract talks about "anticipatory clinical decision making" but the tasks are really about predicting what will be recorded in the EHR, which is different. Predicting that a medication will be charted next hour is not the same as predicting that a patient will deteriorate or benefit from treatment X.

**Requested Changes:**

Be much more honest about the ETHOS comparison. The main table should note that the project compares different feature sets. Either report ETHOS on the full feature space if possible, or clearly separate the analysis into "how much improvement comes from more data vs. better methods." Right now, in my opinion, it's a bit misleading.

Additionally, it will be great to show some qualitative examples of multi-step simulations. Do they look reasonable? Do they drift into nonsensical states after a few iterations? The one example in the appendix is encouraging but author(s) need more analysis. Compare against a simple baseline like "just repeat the last values" to show the LLM simulation is actually better.

Look at performance across patient demographics, across ICU vs. ward vs. ED, across rare vs. common conditions. Does performance collapse for certain groups? This matters for clinical applications.

---

> ### Author Response · Authors · 2026-04-21
> **Official Comment by Authors (Part 1)**
>
> We thank the reviewer for their valuable feedback and for recognizing the “scope of data integration” as “comprehensive”, and “trying to handle the messy reality of what's actually in EHRs”, our “iterative simulation evaluation” as “more interesting than the typical single-step outcome prediction” our “text representation represent\[ing\] clinical semantics reasonably well”. We have added additional clarifications and experiments below and revised the manuscript accordingly. All changes are highlighted in blue in the revised manuscript.
>
> **1\) Fairness and interpretation of the ETHOS comparison:**
> **“The comparison to ETHOS is also problematic. When they restrict to ETHOS-compatible features, the gap shrinks substantially \[...\] The main table should note that the project compares different feature sets. Either report ETHOS on the full feature space if possible, or clearly separate the analysis into "how much improvement comes from more data vs. better methods."”**
>
> **Response:**
>
> We agree that the ETHOS comparison reflects both differences in modeling approach and differences in supported feature space, and that this distinction should be stated more explicitly. We therefore **distinguish between (i) the full pathway-modeling setting**, where broader multimodal input support and broader target coverage are part of the contribution, and **(ii) a matched-output comparison** restricted to ETHOS-supported targets.
>
> In Table 1, we already report ETHOS results separately on the full evaluation space and on the ETHOS-compatible subset, and we now make this distinction explicit in the table caption. **The reduced gap on the shared subset is expected, since EHR2Path is trained to model and predict a broader set of variables**, whereas ETHOS is specialized to a narrower target space. The fact that EHR2Path remains stronger even under the restricted comparison supports the benefit of the proposed approach.
>
> To further disentangle these effects, **we now additionally re-evaluated the hospital lab development task** after restricting both ground truth and predictions to the ETHOS-supported lab vocabulary. This changes the results only minimally (e.g., E2P-S+T Event F1 0.228 → 0.230, ETHOS 0.0465 → 0.0467, with normalized MAE unchanged), because ETHOS already covers many of the most frequent 150 lab labels and almost all evaluated prediction \- ground-truth value pairs fall within this shared lab space. These new results indicate that the remaining gap is not primarily driven by unsupported output labels, but by stronger predictive performance in this task. For the outcome tasks in Table 2 (mortality and discharge), the comparison is already on shared targets. While EHR2Path can use broader input context, we view that as part of the pathway-modeling contribution rather than a separate advantage. We now explicitly discuss this in the revised manuscript.
>
> **Changes in the paper:**
>
> * Revised **Section 5.2 (Patient Trajectory Simulation Results)**: Discuss results in comparison to ETHOS reflect contributions in both increased input space and modelling approach:
>   “\[...\] Because broader multimodal input support is part of the pathway-modeling contribution of EHR2Path, these results reflect both the effect of increased supported input space and differences in modeling approach. \[...\]”
> * Revised **Table 1 caption**: explicitly separate the full-space and ETHOS-compatible comparison:
>   “For ETHOS, results are reported (i) in the full pathway-modeling setting (upper part) and, where outputs ETHOS can not forecast are reflected as negative instances (ii) in an ETHOS-compatible matched-output comparison against E2P-S+T.”
> * **Table 2**: Added footnote stating the minimal changes in results when restricting lab development to ETHOS-supported features.:
>   “Restricting hospital-lab evaluation to the ETHOS-supported lab vocabulary (150 labs) changes results only minimally (\<0.01) and leaves normalized MAE unchanged for both ETHOS and E2P-S+T.”

---

> > ### Author Response · Authors · 2026-04-21
> > **Official Comment by Authors (Part 2)**
> >
> > **2\) Simulation realism and drift:**
> >
> > **“it will be great to show some qualitative examples of multi-step simulations. Do they look reasonable? Do they drift into nonsensical states after a few iterations? The one example in the appendix is encouraging but author(s) need more analysis. Compare against a simple baseline like "just repeat the last values" to show the LLM simulation is actually better.”**
> >
> > **Response:**
> >
> > We agree this is an important topic to consider. We already assess metric behavior over time in Figure 5, showing prediction quality does not degrade substantially across simulated steps. To further improve the evaluation of clinical plausibility in generated trajectories, we strengthened our analysis in two ways:
> >
> > First, we **added an automated clinical plausibility analysis** for generated E2P-T+S rollout windows and compared them to matched real data windows from the test set. The analysis checks four complementary failure modes: values outside the empirical support of the data, implausible one-step jumps, physiologically inconsistent blood-pressure combinations, and unusually frequent medication state changes. Across ED vital signs, ICU vital signs, ICU inputs, hospital medications, and hospital labs, the observed violation rates are low and broadly comparable to the matched real windows. **This new analysis supports that the generated trajectories are not only accurate under task metrics, but also clinically and temporally consistent under these empirical checks.**
> >
> > Second, we **expanded the qualitative multi-step analysis by adding simulations for two feature sets and including copy-forward behavior as an additional visual reference**. This provides an additional qualitative check of whether the learned rollout captures meaningful temporal structure beyond simple persistence and whether it drifts into nonsensical states over multiple steps. We add the following feature sets: 1\) vasopressor combined with blood pressure, showing medication on/off behavior together with a related vital sign, and 2\) a partial hospital renal panel consisting of creatinine and urea nitrogen to illustrate lab value forecasting. The examples show that generated rollouts remain temporally coherent and clinically plausible over multiple steps, without drifting into physiologically nonsensical states. While the model often captures the overall direction of change well, some examples are smoother than the ground truth and miss sharper deviations, especially in the lab setting. In the vasopressor examples, treatment changes are sometimes predicted with a small temporal delay, but the model still captures the correct overall intervention pattern and its coupling to blood-pressure dynamics. We discuss these additional qualitative results in the revised manuscript.
> >
> > **Changes in the paper:**
> >
> > * Revised **Section 5.2 (Patient Trajectory Simulation Results)**: added short description of the main take-aways from the new plausibility analysis
> > * Added **Appendix A.2 \+ Table 7:** Clinical Plausibility Analysis details describing the checks and setup and results.
> > * Extended **Appendix A.6:** additional qualitative results with copy forward comparison
> >
> > New Table 7:
> >
> > Task | Check | Generated | Real
> > -|-|-|-
> > ED vital signs | Range-support violations | 0.47% | 1.07%
> > ED vital signs | One-step jump violations | 0.39% | 1.08%
> > ED vital signs | Blood-pressure consistency violations | 0.00% | 0.00%
> > ICU vital signs | Range-support violations | 0.28% | 1.69%
> > ICU vital signs | One-step jump violations | 0.08% | 0.98%
> > ICU vital signs | Blood-pressure consistency violations | 0.00% | 0.17%
> > ICU inputs | Medication state change outliers | 1.9% | 1.0%
> > Hospital medications | Medication state change outliers | 0.0% | 0.8%
> > Hospital labs | Range-support violations | 0.23% | 1.55%
> > Hospital labs | One-step jump violations | 2.69% | 4.58%
> >
> > **3\) Subgroup analysis:**
> > **“Look at performance across patient demographics, across ICU vs. ward vs. ED, across rare vs. common conditions. Does performance collapse for certain groups?”**
> >
> > **Response:**
> >
> > We agree that performance across different patient groups and clinical settings is important. Robustness across hospital units is already covered by our evaluation design, since tasks are reported separately for **ED, hospital/ward, and ICU** settings. We now make this more explicit in the paper.
> >
> > In addition, we add a targeted subgroup analysis by **age, sex** and **rare/frequent ICD category** to examine whether performance degrades for specific patient groups or rarer clinical conditions. Overall, performance remains broadly stable across the analyzed subgroups, especially for the main trajectory-development tasks and binary outcome predictions, without evidence of systematic degradation in any single subgroup. Age shows a bit larger, but still moderate, heterogeneity, most clearly for ED admission and some value-accuracy metrics, while the ICU development tasks remain comparatively stable.

---

> ### Author Response · Authors · 2026-04-21
> **Official Comment by Authors (Part 3)**
>
> **Changes in the paper.**
>
> * Revised **Section 5.2 (Patient Trajectory Simulation Results)**: Added clarification that the reported simulation results cover **ED, hospital/ward, and ICU** tasks.
> * Added **Appendix A.4 \+ Tables 10/11/12:** Subgroup Heterogeneity Analysis results by sex, age, and icd-level rarity.
>
> New tables without confidence intervals for readability in openreview. Tables in paper include confidence intervals.
>
> Table 10:
>
> Task | Metric | all | male | female
> -|-|-|-|-
> Hospital Discharge Diagnosis | Macro F1 | 0.50 | 0.43 | 0.51
> ED Admission Prediction | Accuracy | 0.68 | 0.67 | 0.69
> ED Admission Prediction | Sensitivity | 0.63 | 0.63 | 0.63
> ED Admission Prediction | Specificity | 0.73 | 0.70 | 0.76
> ED Discharge Diagnosis | Macro F1 | 0.38 | 0.34 | 0.36
> ED Vital Sign Development | Event F1 | 0.57 | 0.58 | 0.56
> ED Vital Sign Development | Value Acc. | 0.51 | 0.50 | 0.51
> ED Vital Sign Development | Value MAE ↓ | 0.10 | 0.10 | 0.10
> Hospital Lab Value Development | Event F1 | 0.23 | 0.23 | 0.22
> Hospital Lab Value Development | Value Acc. | 0.44 | 0.62 | 0.33
> Hospital Lab Value Development | Value MAE ↓ | 0.08 | 0.08 | 0.08
> Hospital Medication Development | Event F1 | 0.83 | 0.83 | 0.82
> ICU Imminent Discharge | Accuracy | 0.69 | 0.68 | 0.70
> ICU Imminent Discharge | Sensitivity | 0.82 | 0.82 | 0.82
> ICU Imminent Discharge | Specificity | 0.56 | 0.55 | 0.59
> ICU Input Development | Event F1 | 0.84 | 0.85 | 0.83
> ICU Imminent Mortality | Accuracy | 0.57 | 0.58 | 0.56
> ICU Imminent Mortality | Sensitivity | 0.16 | 0.17 | 0.16
> ICU Imminent Mortality | Specificity | 0.98 | 1.00 | 0.96
> ICU Vital Sign Development | Event F1 | 0.70 | 0.71 | 0.70
> ICU Vital Sign Development | Value Acc. | 0.79 | 0.79 | 0.79
> ICU Vital Sign Development | Value MAE ↓ | 0.08 | 0.08 | 0.08
>
> Table 11:
>
> Task | Metric | all | young | middle | old
> -|-|-|-|-|-
> Hospital Discharge Diagnosis | Macro F1 | 0.50 | 0.46 | 0.45 | 0.44
> ED Admission Prediction | Accuracy | 0.68 | 0.60 | 0.70 | 0.62
> ED Admission Prediction | Sensitivity | 0.63 | 0.43 | 0.61 | 0.73
> ED Admission Prediction | Specificity | 0.73 | 0.78 | 0.79 | 0.50
> ED Discharge Diagnosis | Macro F1 | 0.38 | 0.40 | 0.30 | 0.35
> ED Vital Sign Development | Event F1 | 0.57 | 0.56 | 0.58 | 0.58
> ED Vital Sign Development | Value Acc. | 0.51 | 0.44 | 0.43 | 0.61
> ED Vital Sign Development | Value MAE ↓ | 0.10 | 0.10 | 0.10 | 0.11
> Hospital Lab Value Development | Event F1 | 0.23 | 0.19 | 0.24 | 0.23
> Hospital Lab Value Development | Value Acc. | 0.44 | 1.00 | 0.00 | 0.42
> Hospital Lab Value Development | Value MAE ↓ | 0.08 | 0.08 | 0.08 | 0.09
> Hospital Medication Development | Event F1 | 0.83 | 0.78 | 0.83 | 0.85
> ICU Imminent Discharge | Accuracy | 0.69 | 0.70 | 0.67 | 0.70
> ICU Imminent Discharge | Sensitivity | 0.82 | 0.86 | 0.79 | 0.79
> ICU Imminent Discharge | Specificity | 0.56 | 0.54 | 0.55 | 0.60
> ICU Input Development | Event F1 | 0.84 | 0.86 | 0.80 | 0.86
> ICU Imminent Mortality | Accuracy | 0.57 | 0.58 | 0.55 | 0.58
> ICU Imminent Mortality | Sensitivity | 0.16 | 0.17 | 0.14 | 0.17
> ICU Imminent Mortality | Specificity | 0.98 | 0.99 | 0.97 | 1.00
> ICU Vital Sign Development | Event F1 | 0.70 | 0.69 | 0.68 | 0.73
> ICU Vital Sign Development | Value Acc. | 0.79 | 0.79 | 0.80 | 0.79
> ICU Vital Sign Development | Value MAE ↓ | 0.08 | 0.08 | 0.08 | 0.08
>
> Table 12:
>
>
> Task | Metric | all | rare condition | common condition
> -|-|-|-|-
> Hospital Discharge Diagnosis | Macro F1 | 0.50 | 0.53 | 0.57
> ED Admission Prediction | Accuracy | 0.68 | 0.72 | 0.65
> ED Admission Prediction | Sensitivity | 0.63 | 0.71 | 0.55
> ED Admission Prediction | Specificity | 0.73 | 0.72 | 0.74
> ED Discharge Diagnosis | Macro F1 | 0.38 | 0.48 | 0.64
> ED Vital Sign Development | Event F1 | 0.57 | 0.57 | 0.58
> ED Vital Sign Development | Value Acc. | 0.51 | 0.47 | 0.54
> ED Vital Sign Development | Value MAE ↓ | 0.10 | 0.10 | 0.10
> Hospital Lab Value Development | Event F1 | 0.23 | 0.19 | 0.27
> Hospital Lab Value Development | Value Acc. | 0.44 | 0.70 | 0.23
> Hospital Lab Value Development | Value MAE ↓ | 0.08 | 0.08 | 0.08
> Hospital Medication Development | Event F1 | 0.83 | 0.84 | 0.81
> ICU Imminent Discharge | Accuracy | 0.69 | 0.69 | 0.69
> ICU Imminent Discharge | Sensitivity | 0.82 | 0.83 | 0.81
> ICU Imminent Discharge | Specificity | 0.56 | 0.55 | 0.57
> ICU Input Development | Event F1 | 0.84 | 0.85 | 0.84
> ICU Imminent Mortality | Accuracy | 0.57 | 0.60 | 0.55
> ICU Imminent Mortality | Sensitivity | 0.16 | 0.19 | 0.13
> ICU Imminent Mortality | Specificity | 0.98 | 1.00 | 0.97
> ICU Vital Sign Development | Event F1 | 0.70 | 0.70 | 0.70
> ICU Vital Sign Development | Value Acc. | 0.79 | 0.78 | 0.80
> ICU Vital Sign Development | Value MAE ↓ | 0.08 | 0.08 | 0.08

---

> > ### Author Response · Authors · 2026-04-21
> > **Official Comment by Authors (Part 4)**
> >
> > **4\) Novelty and Analysis of the summarization bottleneck**
> >
> > **“The summarization bottleneck is basically just a custom attention mask that forces outputs to attend only to summary tokens. This is intuitive but not particularly novel. It's unclear why this is better than just using established efficient attention mechanisms. \[...\] We don't know if the summary tokens are capturing meaningful clinical patterns or just fitting whatever random initialization works. There's no visualization, no analysis of what gets compressed away. They don't compare to simpler alternatives like just truncating history or using sparse attention.”**
> >
> > **Response:**
> > With respect to the summarization bottleneck, the **key contribution lies in the combination of the section-wise summarization mechanism and the forecasting objective used to learn the summary tokens**. Together, these enable step-wise compression of very long inputs while explicitly encouraging the compressed representation to retain information that is useful for forecasting.
> > Regarding simpler alternatives, **E2P-T (1h) and E2P-T (24h) already serve as truncation-style baselines** that allocate the available budget entirely to recent raw text, based on the assumption that general patient context together with the most recent information is most relevant for forecasting. In contrast, **E2P-S tests whether the learned summary representation retains useful predictive information on its own** through downstream task performance. The fact that the summary-only model remains clearly predictive suggests that the summary tokens capture meaningful signals.
> >
> > Lastly, we **added a qualitative analysis of the learned summary-token attention** to provide additional evidence that the bottleneck captures meaningful, forecast-relevant structure rather than arbitrary compressed features. For four representative section types (procedures, medications, vital signs, and lab results), we visualize heatmaps showing which structured clinical input spans, including both field names and associated values, receive the strongest attention from each of the eight summary tokens. We additionally indicate which span topics reappear in the next-hour output. For readability, each heatmap shows the union of the top-5 attended spans across tokens.
> > The visualizations show two key patterns. First, different summary tokens focus on different parts of the input, suggesting that the compressed representation is distributed across tokens rather than collapsing onto the same spans. Second, the dominant attended spans frequently overlap with information that appears in the next-hour output, indicating that the bottleneck emphasizes forecast-relevant content, which is exactly what the summary-model training objective is designed to encourage. Taken together, **these results support that the bottleneck learns a non-trivial, task-aligned compression.**
> >
> > **Changes in the paper.**
> >
> > * Revised **Section 5.1 (Next Timestep Prediction Results):** position **E2P-T (1h)** and **E2P-T (24h)** as recent-text-only truncation baselines under the same practical input budget and **E2P-S** provides evidence that the learned summary tokens retain meaningful predictive signal on their own.
> > * Added **Appendix A.7**: Attention Visualizations of learned summary tokens
> >
> > **5\) Representation compactness / token efficiency**
> >
> > **“The structured text representation makes sense but it's verbose. An average hour is 380 tokens which adds up. Did you consider more compact representations? Is this formatting actually efficient for the LLM or are you wasting tokens?”**
> >
> > **Response:**
> >
> > The **main source of this verbosity is the underlying EHR density**. A central goal of EHR2Path is broad coverage of heterogeneous multimodal patient state rather than reducing the input to a small selected variable set, and this necessarily trades some token efficiency for more complete state representation. Even a 1-hour ICU window can contain a large number of simultaneously charted variables, and our serializer preserves this high-dimensional state explicitly. The current format already avoids serializing missing variables and compresses repeated values within each variable. **Additionally, the Masked Summarization Bottleneck was introduced precisely to address this challenge** at the architectural level, by compressing long-history context into compact summary tokens under a fixed input budget. We therefore view the **current representation as a trade-off between compactness and coverage, and we now make this trade-off explicit in the paper.**
> >
> > **Changes in the paper:**
> >
> > * Revised **Section 3.1 (Data Representation):** Added discussion that **broad multimodal coverage** is a central design goal of the representation, but leads to increased representation size.
> > * Revised **Section 5.1 (Next Timestep Prediction Results):** Explicitly discusses that broad coverage can lead to higher token cost even for short time windows.

---

> > > ### Author Response · Authors · 2026-04-21
> > > **Official Comment by Authors (Part 5)**
> > >
> > > **6\) Scope and framing**
> > >
> > > **“The experimental validation is narrow, restricted entirely to MIMIC-IV. For a paper claiming ‘scalable’ modeling this is a significant limitation. \[...\] The authors never test on any other healthcare system, patient population, or EHR structure. \[..\] The paper also needs to answer whether clinicians care about these predictions or would use them. The abstract talks about ‘anticipatory clinical decision making’ but the tasks are really about predicting what will be recorded in the EHR, which is different. Predicting that a medication will be charted next hour is not the same as predicting that a patient will deteriorate or benefit from treatment X.”**
> > >
> > > **Response:**
> > >
> > > We agree the current validation, limited to MIMIC-IV, does not establish transferability across healthcare systems, which is why **we acknowledged this in the limitation section of the original manuscript**. At the same time, we believe the current evaluation remains valuable because MIMIC-IV is a large, heterogeneous, and clinically realistic benchmark, allowing us to test the feasibility of pathway-level modeling under broad multimodal and long-context conditions.
> > > We agree the framing can be refined in some places and have r**evised the manuscript to directly address the mentioned claims**:
> > >
> > > - With **‘scalable’** we were referring to the amount, heterogeneity, and temporal extent of the considered EHR data, rather than demonstrated scalability across healthcare systems, but we understand this term is ambiguous and **reformulated it as “pathway-level”.**
> > > - We further **narrowed our formulations regarding ‘anticipatory clinical decision making’** to more clearly reflect our contribution: We show the feasibility of pathway-level modeling on long, heterogeneous, multimodal in-hospital trajectories within a real-world EHR dataset and study retrospective patient-state forecasting and trajectory simulation from structured EHR. At the same time, we believe such models can be clinically relevant as tools for monitoring, simulation, and hypothesis generation in the future, with any decision-support use requiring external and prospective validation, which we now discuss explicitly as future work.
> > > - We also **revised the Broader Impact Statement** accordingly. It now clarifies that our results do not establish readiness for real-world decision support, and that any clinical use would require use-case-specific validation, privacy-preserving data handling, appropriate governance, and clinician oversight.
> > >
> > > **Changes in the paper:**
> > >
> > > - We adapted the **title, abstract and conclusion** to better reflect our intended scope
> > > - We **changed the word “scalable”** throughout the paper to “pathway-level” in order to clearly show that we refer to scalability with respect to the amount, heterogeneity, and temporal extent of the considered EHR data, rather than demonstrated scalability across healthcare systems.
> > > - Revised **Section 5.5 (Limitations and Future Work):** Added explicit limitation that broader clinical use would require external and prospective validation.
> > > - Revised the **Broader Impact Statement** accordingly.
> > >
> > > **7\) Figure 4 clarity**
> > >
> > > **“Figure 4 summarizing task coverage is nice but actually hard to interpret because you're mixing task types. ED diagnosis prediction is a different kind of task than vital sign forecasting. Mixing them in one visualization obscures more than it reveals.”**
> > >
> > > Thank you for this helpful suggestion. To improve interpretability, **we revised Figure 4 to separate the different task types** into separate radar plots, rather than combining them in a single visualization. In particular, we now show (i) simulation-based development tasks, (ii) simulation-based outcome tasks, and (iii) fine-tuned outcome tasks separately.
> > >
> > > **Changes in the paper:**
> > >
> > > * Adapted **Figure 4**
> > >
> > > **8\) Privacy / deployment considerations**
> > >
> > > **“I encourage the author(s) to discuss privacy considerations for anyone who wants to deploy or adapt this to real hospitals. What validation do they need to do? What are they responsible for?”**
> > >
> > > **Response:**
> > >
> > > Thank you, this is a helpful suggestion. We **added a discussion of deployment considerations for adapting pathway trajectory models to real hospital settings**. In particular, we now clarify that deployment of such models would require validation on the target hospital’s data and intended use case, including subgroup and workflow-specific evaluation, and that deployers are responsible for privacy-preserving data handling, appropriate governance and access controls, post-deployment monitoring, and clinician oversight. We also note that the model can be run locally, supporting privacy considerations.
> > >
> > > **Changes in the paper:**
> > >
> > > * Revised **Section 5.5 (Limitations and Future Work):** Added discussion on deployment considerations

---

### Review · Reviewer_Yama · 2026-04-28

**Summary Of Contributions:**

This paper introduces EHR2Path, a multimodal framework for forecasting and simulating in-hospital patient pathways from routine EHRs. It uses a unified textual representation of heterogeneous clinical data and a Masked Summarization Bottleneck to compress long patient histories into summary tokens. Experiments on MIMIC-IV show that EHR2Path supports next-step prediction, iterative trajectory prediction, and outcome prediction across ED, Hospital, and ICU tasks.

**Audience:**

Yes

**Audience Explanation:**

Yes. The paper would likely be of interest to part of the TMLR audience, particularly researchers working on clinical AI and multimodal long-context health data. The idea of moving beyond isolated outcome prediction toward pathway-level forecasting and iterative simulation is timely and relevant. The proposed Masked Summarization Bottleneck may also be interesting to readers studying efficient long-context modeling.

**Claims And Evidence:**

Yes

**Claims Explanation:**

Overall, the main claims are generally supported by the experiments. The submission provides clear evidence for next-step pathway forecasting on MIMIC-IV and shows that EHR2Path can model a broad set of heterogeneous EHR data. It also provides evidence for iterative simulation, improved context efficiency through the Masked Summarization Bottleneck, and outcome prediction performance.
However, the baseline comparisons are not fully controlled, since EHR2Path supports a broader feature space than the compared methods. As a result, some of the observed gains may come from broader data integration rather than the proposed modeling approach alone.

**Requested Changes:**

1. Could the authors clarify how positive and negative labels are determined from multi-step rollouts, what stopping rule is used, and whether any decision thresholds are selected on the validation set and then fixed for testing?
2. Could the authors discuss whether the model may partly learn documentation and measurement patterns rather than only patient-state evolution, since the sparse output includes only features actively recorded at the next time point?
3. The evaluation is limited to MIMIC-IV, so it remains unclear how well EHR2Path generalizes to other hospitals, documentation practices, and patient populations.
4. The paper shows that adding LOS improves convergence for discharge-related tasks. Could the authors further analyze how errors in the predicted LOS affect rollout termination?

---

> ### Author Response · Authors · 2026-04-30
> **Official Comment by Authors (Part 1)**
>
> We thank the reviewer for their positive feedback and for recognizing the “idea of moving toward \[...\] pathway-level forecasting as timely and relevant”, and our experiments as providing “clear evidence for next-step pathway forecasting on MIMIC-IV”. We have added additional clarifications and experiments below and revised the manuscript accordingly. All changes are highlighted in blue in the revised manuscript.
>
> **1\) Could the authors clarify how positive and negative labels are determined from multi-step rollouts, what stopping rule is used, and whether any decision thresholds are selected on the validation set and then fixed for testing?**
>
> **Response:**
>
> Our model generates a structured textual rollout, and prediction labels are extracted by rule-based parsing of generated disposition, death, and release events. Therefore, no model probabilities are used to obtain predictions, and no decision threshold is selected on the validation set.
>
> For ED admission, the ground-truth label is positive if the final ED disposition is hospital admission. The rollout-derived prediction is positive if the generated ED disposition is ADMITTED, and negative if the generated terminal ED disposition is DISCHARGED or DIED. The rollout stops when an ED terminal disposition or admission transition is generated. We also use a fixed maximum-step guard, which would stop the simulation and assign a negative label. However, this guard is chosen large enough that we did not observe it being reached in practice.
>
> For ICU imminent mortality, the ground-truth label is positive if death occurs within the 24-hour prediction window. The rollout-derived prediction is positive if the generated trajectory contains a death disposition within this horizon, and negative otherwise. The rollout stops at death/discharge or at the fixed 24-hour horizon.
>
> For ICU imminent discharge, the ground-truth label is positive if ICU release occurs within the 72-hour prediction window. The rollout-derived prediction is positive if an ICU release event is generated within 72 hours, and negative otherwise. The rollout stops at ICU release/death or at the fixed 72-hour horizon.
>
> We have adapted the paper to describe these rollout definitions in more detail.
>
> **Changes in the paper:**
>
> \- Extended **Appendix A.9 (Detailed Task Descriptions)**:
> \- Added above explanations into task definitions and general information about rollout termination.
> \- Clarified that labels are obtained by rule-based parsing of generated structured outputs and rollouts terminate when terminal state or maximum horizon is reached.
>
> **2\) Could the authors discuss whether the model may partly learn documentation and measurement patterns rather than only patient-state evolution, since the sparse output includes only features actively recorded at the next time point?**
>
> This is an important point. EHR2Path predicts the next recorded EHR state, and therefore the learned signal indeed combines patient-state evolution with hospital measurement and documentation patterns. This is partly inherent to the task. Whether a lab is ordered, a medication is administered, or a vital sign is charted is itself informative about the clinical workflow and tests or interventions considered necessary at a given time point. At the same time this means the model should not be interpreted as recovering a fully observed physiological trajectory independent of documentation practices, as sparse outputs indicate what is expected to be recorded next, not necessarily all clinically true but unmeasured patient states. We now added an explicit discussion of this point in the manuscript.
>
> **Changes in the paper:**
>
> - Section **5.5 Discussion and Limitations:** Added above discussion.
>
> **3\) The evaluation is limited to MIMIC-IV, so it remains unclear how well EHR2Path generalizes to other hospitals, documentation practices, and patient populations.**
>
> **Response:**
>
> We agree that validation on MIMIC-IV alone does not establish generalizability across hospitals, documentation practices, or patient populations, which is why **we acknowledged this in the limitation section of the original manuscript**. We have **also revised the framing**, clarifying that the paper demonstrates retrospective pathway modeling on long, multimodal MIMIC-IV trajectories. At the same time, we believe the current evaluation remains valuable because MIMIC-IV is a large, heterogeneous, and clinically realistic benchmark, allowing us to test the feasibility of pathway-level modeling under broad multimodal and long-context conditions.
>
> In addition, while this does not replace multi-dataset validation, during the revision we added a subgroup analysis within MIMIC-IV (Appendix A.4) by age, sex, and ICD group rarity showing that there are no major within-dataset performance degradation across these patient groups.

---

> > ### Author Response · Authors · 2026-04-30
> > **Official Comment by Authors (Part 2)**
> >
> > **Changes in the paper:**
> >
> > - We adapted the **title, abstract and conclusion** to better reflect our intended scope
> > - We **changed the word “scalable”** throughout the paper to “pathway-level” in order to clearly show that we refer to scalability with respect to the amount, heterogeneity, and temporal extent of the considered EHR data, rather than demonstrated scalability across healthcare systems.
> > - Revised **Section 5.5 (Limitations and Future Work):** Discuss external and prospective validation as important future work.
> > - Added **Appendix A.4 \+ Tables 10/11/12:** Subgroup Heterogeneity Analysis results by sex, age, and icd-level rarity.
> >
> > **4\) The paper shows that adding LOS improves convergence for discharge-related tasks. Could the authors further analyze how errors in the predicted LOS affect rollout termination?**
> >
> > **Response:**
> >
> > The relevance of LOS indicator behavior differs across the two discharge-related tasks in the LOS ablation. For **Hospital Discharge Diagnosis**, the LOS indicator is set to 0 in the input to trigger direct discharge prediction. In this setting, the **model converged to a discharge prediction in all evaluated cases**, consistent with the 100% convergence rate reported in the ablation table. Thus, **LOS prediction errors do not affect the results** for this task in the same way as in iterative rollout termination.
> >
> > For **ICU Imminent Discharge**, termination behavior is directly relevant because the rollout-derived label depends on whether ICU release is generated within the 72-hour horizon. We therefore **decomposed the test set predictions into termination-outcome categories**:
> >
> > | Category | Count | Percentage |
> > |---|---:|---:|
> > | Correct imminent discharge | 204 | 40.80% |
> > | Correct non-imminent discharge | 141 | 28.20% |
> > | Premature predicted discharge | 109 | 21.80% |
> > | Missed/delayed discharge | 46 | 9.20% |
> >
> > The above table shows that the **majority of rollouts fall into the two correct termination categories**, while incorrect termination can be separated into premature predicted discharge and missed/delayed discharge, with premature discharge being the more common error mode. Such **errors can happen when the initial LOS indicator predicted in the first step already underestimates or overestimates the remaining stay, or when it decreases too quickly, stalls, or is re-estimated incorrectly during rollout.**
> >
> > To further illustrate how LOS-indicator dynamics influence termination across all four categories, **we added a qualitative visualization of LOS trajectories for one correct imminent, correct non-imminent, premature, and missed/delayed discharge case each**. These examples show how the LOS indicator is re-estimated during rollout, allowing the model to update its remaining-stay estimate based on the evolving predicted pathway.Therefore, both the initial LOS estimate and its subsequent trajectory can affect whether a discharge event is generated before the 72-hour horizon. Overall, the trajectories mostly remain in an appropriate LOS range even when the exact remaining stay length is difficult to predict, such as in stays where discharge is still far in the future.
> >
> > **Changes in the paper:**
> >
> > * Revised **Section 5.4 Additional Ablation Results**:
> >   * Added **new Table 4**: overall termination-outcome analysis
> >   * Added **Figure 6**: qualitative LOS indicator rollout examples
> >   * Interpretation of new analysis
> >
> > **5\) EHR2Path supports a broader feature space than the compared methods. As a result, some of the observed gains may come from broader data integration rather than the proposed modeling approach alone.**
> >
> > **Response:**
> >
> > We agree that baseline comparisons reflect both differences in modeling approach and differences in supported feature space. EHR2Path is designed to model a broader multimodal patient state, and this broader input and output coverage is a part of the pathway-modeling contribution. At the same time, we agree that the results should not be attributed to architectural contributions alone. We therefore revised the manuscript to state this explicitly when discussing baseline comparisons.
> >
> > **Changes in the paper:**
> >
> > * Revised **Section 5.2 (Patient Trajectory Simulation Results)**: Discuss explicitly that results reflect contributions in both increased input space and modelling approach:
> >   “\[...\] Because broader multimodal input support is part of the pathway-modeling contribution of EHR2Path, these results reflect both the effect of increased supported input space and differences in modeling approach. \[...\]”
> > * Revised the **Table 1 caption**: More explicitly separate the full pathway-modeling setting from the ETHOS-compatible matched-output comparison.

---

> > > ### Comment · Reviewer_Yama · 2026-05-16
> > > **Response to rebuttal**
> > >
> > > Thank you for the detailed response and revisions.
> > >
> > > The clarifications on the simulation protocol and the additional LOS termination analysis satisfactorily address my main concerns, although I still view external generalization and fully controlled baseline comparisons as remaining limitations. Given that premature predicted discharge occurs in 21.8% of ICU imminent discharge cases, could the authors explicitly discuss this as a limitation?

---

> > > > ### Author Response · Authors · 2026-05-18
> > > > **Response to Reviewer**
> > > >
> > > > Thank you for the positive feedback. We have revised Section 5.4 to explicitly discuss the 21.8% premature predicted discharge as a limitation of the current LOS-guided rollout procedure, leading to sometimes generating ICU releases too early relative to the 72h decision window of the LOS task. Please let us know if you have any further comments or questions.

---

> > > > > ### Comment · Reviewer_Yama · 2026-05-18
> > > > >
> > > > > Yes, I see. Thanks for addressing my comments.

---

### Comment · Action_Editor_rvxM · 2026-05-10
**Reviewers: please respond to author rebuttal**

To reviewers:
Thank you for your reviews. The authors have posted detailed responses and updated the paper in light of your feedback. Please engage with the authors' rebuttal so the discussion can inform the recommendation phase.
--AE